# Head Pursuit: Probing Attention Specialization in Multimodal Transformers

**Lorenzo Basile**[1,*]    **Valentino Maiorca**[2,3]    **Diego Doimo**[1]

**Francesco Locatello**[3]    **Alberto Cazzaniga**[1,*]

[1]Area Science Park, Italy
[2]Sapienza University of Rome, Italy
[3]Institute of Science and Technology, Austria

## Abstract

Language and vision-language models have shown impressive performance across a wide range of tasks, but their internal mechanisms remain only partly understood. In this work, we study how individual attention heads in text-generative models specialize in specific semantic or visual attributes. Building on an established interpretability method, we reinterpret the practice of probing intermediate activations with the final decoding layer through the lens of signal processing. This lets us analyze multiple samples in a principled way and rank attention heads based on their relevance to target concepts. Our results show consistent patterns of specialization at the head level across both unimodal and multimodal transformers. Remarkably, we find that editing as few as 1% of the heads, selected using our method, can reliably suppress or enhance targeted concepts in the model output. We validate our approach on language tasks such as question answering and toxicity mitigation, as well as vision-language tasks including image classification and captioning. Our findings highlight an interpretable and controllable structure within attention layers, offering simple tools for understanding and editing large-scale generative models.

## 1   Introduction

Large-scale generative models, including both language and vision-language transformers, have achieved remarkable performance on a wide spectrum of tasks, from open-ended text generation [1] to image captioning and visual question answering [2–5]. Despite these successes, the internal mechanisms by which these models organize and represent knowledge are still incompletely understood. In particular, the role of individual components, such as attention heads, in mediating specific aspects of generation has been the subject of increasing interest for both interpretability and control [6–8]. Previous studies have shown that attention heads in large language models (LLMs) often exhibit emergent roles, such as syntax tracking or copy behavior [9–11]. Interpretability tools such as the Logit Lens [12] and its extensions [13, 14] have provided strategies for inspecting intermediate model representations, revealing rich semantic information latent in hidden states. However, these techniques are typically applied heuristically and focus on individual examples, making it difficult to generalize findings across multiple samples or quantify the importance of specific model components in shaping the model's output.

In this work, we take a more principled approach to analyzing the specialization of attention heads in generative transformers. The foundation of our approach is to reinterpret existing interpretability

---

*Correspondence to: {lorenzo.basile, alberto.cazzaniga}@areasciencepark.it

39th Conference on Neural Information Processing Systems (NeurIPS 2025).

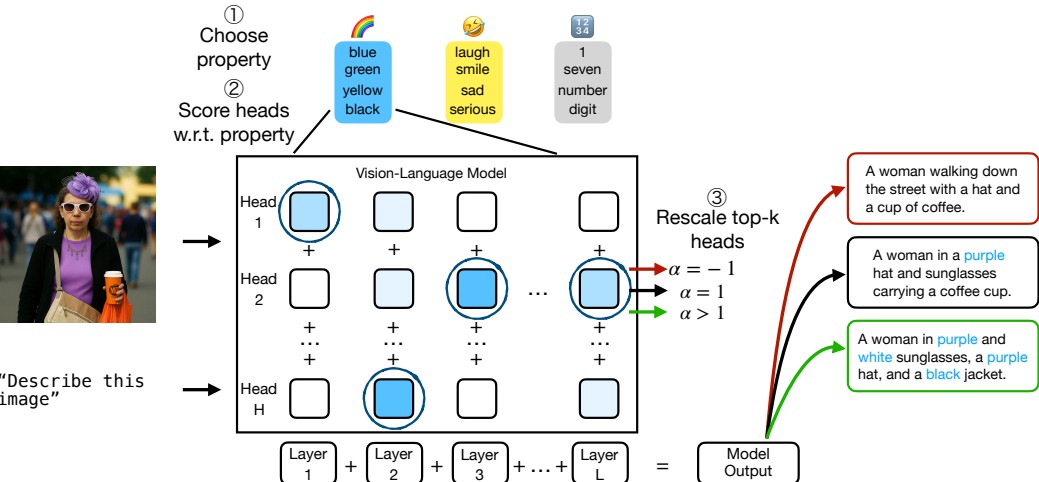

Figure 1: Overview of our method. Given a language or vision-language model and a target property defined by text (e.g., *colors*, *sentiments*, *numbers*), we score all attention heads according to how well they align with interpretable directions from a fixed dictionary, using a method based on Matching Pursuit [15] (1). We then select the top-$k$ heads (2) and intervene by rescaling their contribution to the residual stream (3), either enhancing or suppressing the attribute in the model's output.

tools through the lens of sparse signal recovery. Specifically, we revisit a variant of Matching Pursuit (MP) [15], a classical greedy algorithm to approximate high-dimensional signals with sparse linear combinations of basis elements, and bridge it with recent interpretability techniques. By applying MP to the hidden states of text-generative models, we propose a way to identify a small set of attention heads that most strongly influence the model's capability to generate text within specific conceptual domains (e.g., *colors* or *numbers*). This approach provides a mathematically grounded strategy to decompose model behavior into a small set of interpretable elements, contrasting with prior heuristic techniques, thus enabling both quantitative analysis and targeted interventions on model mechanisms.

Using this framework, we reveal consistent head specialization patterns across large unimodal and multimodal pre-trained models. We find that certain heads are reliably responsible for generating semantically coherent groups of tokens, such as names, colors, or sentiment-bearing words. Moreover, we find that intervening on just a small fraction of these concept-specific heads can significantly affect the model's output, enabling suppression and enhancement of targeted content. These results suggest that attention layers contain a highly interpretable and manipulable linear structure, consistent with evidence that large-scale models represent high-level concepts in approximately linear residual subspaces [16–19]. Such structure opens new directions for understanding and controlling model behavior without additional training. In particular, the emergence of structured, concept-aligned subspaces indicates that a small number of units can exert broad semantic control over generation, offering a principled alternative to heuristic prompt-based interventions and highlighting the potential of sparse, head-level manipulation as a general tool for studying and shaping internal model behavior.

Our contributions can be summarized as follows:

- We introduce a strategy to frame Matching Pursuit (MP), an established sparse recovery algorithm, in the context of the interpretability of generative language models, establishing its connection to standard interpretability tools such as the Logit Lens;

- We apply MP to reveal that attention heads of LLMs often specialize in the generation of tokens belonging to narrow semantic areas, and propose an approach to identify the group of heads most relevant for a conceptual domain;

- We demonstrate that head specialization opens up a way to manipulate model behavior. Both in language and multimodal tasks, negating relevant heads causes targeted degradation in task performance, while enhancing them promotes the generation of specific attributes.

## 2 Related work

Recent research on Transformer architectures has investigated the functional roles and specialization of attention heads. In language models, most heads appear redundant, with pruning studies showing that many can be removed with minimal loss in performance on NLP tasks. Only a few contribute significantly to linguistic functions such as encoding positional information, syntactic structure, or attending to rare words [9]. Some heads have also been linked to eliciting factual knowledge [6], promoting in-context induction [11], or suppressing lexical repetition [20]. Other analyses have examined attention heads through their learned weights [21], providing insight into how information is routed within the model. Further work has explored the localization and manipulation of MLP and residual representations. Early mechanistic interpretability studies [22, 23] showed that factual associations are encoded primarily in mid-layer MLPs and can be modified by targeted intervention on MLP weights.

In vision-language models, similar specialization patterns have been observed in the visual encoder of CLIP-like architectures by leveraging visual-textual alignment to decompose heads over sentence encodings [24, 25]. Beyond contrastive models, recent work has applied dictionary learning to generative VLMs to extract human-interpretable concepts from latent activations [26, 27], building on earlier efforts in CNN interpretability [28]. A parallel line of research adapts mechanistic interpretability tools from language models to the multimodal setting. Representative works include [29] and [30], which investigate information transfer mechanisms in multimodal transformers, and [31], which extends the Logit Lens [12] to the analysis of visual token representations.

Closely related to our approach are methods derived from the Logit Lens (LL) [12], which interpret internal representations by projecting them into the output space. The Attention Lens [14] extends LL to individual attention heads but requires training a separate linear probe for each head, following the Tuned Lens framework [13], making it computationally expensive and difficult to scale to billion-parameter models. In contrast, our method identifies concept-relevant heads by directly projecting their outputs onto the model's unembedding matrix, without requiring any additional training. Another related method [32] also employs the unembedding matrix for interpretability, but within a gradient-based saliency framework that attributes model predictions to influential input tokens. While their approach is task-dependent and gradient-based, ours is task-agnostic and operates purely on model representations, and the two could be combined to focus our analysis on salient tokens.

In summary, our work provides a unifying perspective across these directions by investigating head specialization in generative language and vision-language models through sparse decomposition over a fixed dictionary of interpretable directions. Rather than learning the dictionary from activations, we assume a known semantic basis, typically derived from the unembedding matrix, and use sparse recovery to identify heads whose outputs align with specific attributes.

## 3 Pursuing specialized attention heads

We start our investigation by exploring whether individual attention heads of generative LLMs specialize in interpretable functions. To isolate the contribution of each head, we use a residual stream decomposition approach. This allows us to assess how each attention head contributes to the residual stream at a head-level granularity. Specifically, following [33], we model the output written by each head into the residual stream as a matrix $\mathbf{H}_{h,l} \in \mathbb{R}^{n,d}$, where $n$ is the number of samples in the dataset and $d$ is the internal dimensionality of the transformer.

Motivated by the growing body of literature that uses latent decompositions for interpretability, especially in vision-language models [24–27], our aim is to identify sparse and interpretable directions for each attention head $\mathbf{H}_{h,l}$ that maximally explain its variance on a given dataset. Concretely, we seek a sparse representation of $\mathbf{H}_{h,l}$ using directions drawn from a fixed dictionary of interpretable vectors, rather than an unconstrained continuous space, to ensure that the resulting components are meaningful and grounded in known semantic structures.

As a dictionary, we adopt the unembedding matrix of the language model $\mathbf{D} \in \mathbb{R}^{v,d}$, as it naturally contains directions that are aligned with semantically meaningful outputs, allowing us to ground latent structure in human-interpretable terms. In fact, every row of this matrix is a $d$-dimensional vector that effectively represents in the latent space a token that can be decoded into natural language.

We then construct an approximation of each head representation using directions from our dictionary (i.e., the unembedding matrix) via a classical sparse coding algorithm: Simultaneous Orthogonal Matching Pursuit (SOMP) [34] (see Appendix A). SOMP is a multi-sample extension of Orthogonal Matching Pursuit [35], itself a refinement of the original Matching Pursuit algorithm [15]. Rather than analyzing each sample independently, SOMP jointly considers all samples in a given dataset and selects the dictionary directions that are most informative across the representation.

Formally, given a head activation matrix $\mathbf{H} \in \mathbb{R}^{n,d}$ and a dictionary $\mathbf{D} \in \mathbb{R}^{v,d}$, SOMP aims to iteratively construct a column-sparse coefficient matrix $\mathbf{W}^* \in \mathbb{R}^{n,v}$ such that:

$$\mathbf{H} \approx \mathbf{W}^*\mathbf{D} \tag{1}$$

At each iteration $t$, the algorithm selects the dictionary atom (i.e., a row of $\mathbf{D}$) that maximally correlates with the head residuals across all samples:

$$p^t = \arg\max_j \left\| \mathbf{D}[j]\mathbf{R}^{t^T} \right\|_1 \tag{2}$$

Here, the head residual matrix $\mathbf{R}^t \in \mathbb{R}^{n,d}$ is defined as the difference between the original signal and its reconstruction at step $t$: $\mathbf{R}^t = \mathbf{H} - \mathbf{H}_r^t$. The selected index $p^t$ is added to the support set $\mathbb{S}^{t+1}$, and the dictionary is refit by solving a least-squares problem restricted to the current support:

$$\mathbf{W}^t = \arg\min_{\mathbf{W}} \left\| \mathbf{H}_{h,l} - \mathbf{W}\mathbf{D}[\mathbb{S}^{t+1}] \right\|_F \tag{3}$$

The reconstruction is updated as $\mathbf{H}_r^{t+1} = \mathbf{W}^t\mathbf{D}[\mathbb{S}^{t+1}]$, and the residuals are recomputed accordingly. This iterative process continues until a predefined sparsity level is reached. The resulting decomposition expresses each head's output using a sparse set of semantically meaningful dictionary atoms, yielding an interpretable approximation of its behavior.

Importantly, we note a conceptual connection between our reinterpretation of SOMP and the Logit Lens (LL) [12], a tool widely used in mechanistic interpretability to probe internal representations of transformer models. Similarly to the method just described, LL works by projecting a single residual stream vector onto the unembedding directions to approximate the output logits of the model at intermediate layers. This is equivalent to performing a single step of Matching Pursuit on an individual example. Our SOMP-based method generalizes this idea in two key ways: it operates on multiple examples simultaneously, and it selects multiple dictionary directions, each capturing distinct components of the signal. This leads to a more robust and semantically structured characterization of the attention head's functional role.

In Table 1, we report some examples of specialized attention heads, obtained by applying SOMP to Mistral-7B attention heads, prompted by questions from the TriviaQA dataset [36]. Before applying SOMP, the tokens from the prompt were aggregated by averaging. As we show in Table 6, a direct application of LL in this setting results in noisier and highly redundant explanations.

Table 1: Top-5 tokens identified by SOMP on selected attention heads of Mistral-7B, evaluated on TriviaQA prompts.

| L18.H27 ("Politics") | L24.H20 ("Nationality") | L25.H14 ("Months") | L30.H28 ("Numbers") |
|---|---|---|---|
| COVID | British | December | 9 |
| Soviet | American | July | 1 |
| Obama | European | April | 3 |
| Biden | German | October | 7 |
| Clinton | English | February | five |

Besides returning lists of latent directions and associated natural language tokens that better characterize each head, SOMP produces a reconstruction of the head representation in the space spanned by those vectors. Building on this insight, we propose a method to automatically identify the heads most relevant for a target attribute. Given a list of words related to the chosen semantic area, one can restrict the unembedding matrix to the rows associated to these tokens and apply SOMP on this concept-specific dictionary. Then, the fraction of head variance explained by SOMP in this setting can be considered as a measure of specialization of the head, allowing us to rank and select heads by their relevance with respect to the target concept.

# 4 Controlling language generation through specialized heads

We now evaluate how the specialization of attention heads can be leveraged to apply domain-specific targeted interventions to model behavior, effectively validating our selection. One way to do so is to disrupt the information flow from a selected subset of heads to the residual stream during the forward pass. Concretely, we apply this intervention by inverting the sign of the head representations. It is worth noting that this intervention not only affects the direct contribution of the head to the residual stream, but also the indirect contribution of its information content to subsequent layers.

The key preliminary step is to identify relevant and specialized heads. To accomplish this, we apply SOMP (see Section 3) over a restricted unembedding dictionary, filtered to include only a set of tokens associated with the target property. These tokens can be selected using various strategies, such as user-defined word lists, class names, or keywords generated by an external LLM. Then, we rank attention heads by the proportion of variance in the data explained by SOMP (in our experiments, typically using 50 iterations) and intervene on the top-$k$ ranked heads.

In all of our experiments, we include a random control condition to verify the specificity of our findings. This control involves intervening on a randomly selected set of attention heads that matches the original set in both size and layer distribution but is entirely disjoint from it. For these purposes, to ensure a fair comparison, we explicitly avoid selecting heads previously identified as specialized when constructing the control set. In all experiments, we report such random control results over 10 independently sampled sets of heads.

## 4.1 Question answering

**Experimental setting** We consider a generative LLM, Mistral-7B [37], and evaluate it on textual prompts from the TriviaQA [36] question answering dataset. Our goal is to identify attention heads specialized in generating country names, a target attribute motivated by their relative abundance in the dataset: despite their specificity, country names account for over 6% of the answers in the test split. For targeting the countries concept, we restrict the tokens in our dictionary (unembedding matrix) to those corresponding to names of countries, and apply our Matching Pursuit based method to select specialized heads. As an additional baseline, we report results obtained by inverting heads selected with a simple adaptation of the Logit Lens. Specifically, we score each head with the mean logit assigned to country-related tokens by LL, and select top-$k$ heads as in our method. Importantly, head representations are computed using questions from the training data, which is strictly disjoint from the data used in evaluation. Model performance on TriviaQA is assessed using the standard F1 score, which accounts for partial overlaps between the predicted and ground-truth answers.

**Result analysis** We report the results of our intervention on subsets of specialized heads in Figure 2, varying the number of selected heads. Performance on the target attribute (solid blue line) noticeably degrades when the signs of 8 heads (0.8% of the total) or more are inverted. Notably, performance on the remaining examples (solid orange line) declines more gradually, suggesting that these specialized heads have a targeted effect. Since the semantic domains of the questions answered by a country name and the remaining part of TriviaQA are not disentangled, it is expected that intervening on the selected heads also has a (lower) impact on the remaining examples (orange line). Control experiments using random heads (faded lines) show no significant impact on performance, confirming the specificity of the identified heads. Instead, heads selected by the Logit Lens (dashed lines) are relevant to the question-answering tasks but not specific to the targeted concept, as they degrade performance equally within and outside the targeted domain.

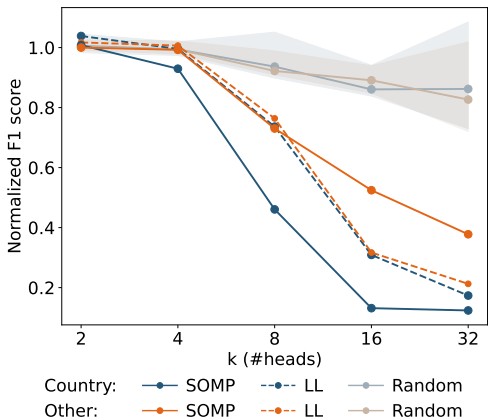

Figure 2: Question answering performance of Mistral-7B on TriviaQA. F1 scores are shown for samples associated (country) or not (other) with the target attribute, normalized to the base model accuracy without intervention. Random baselines are displayed as medians and interquartile ranges.

Overall, the analysis suggests that our method correctly identifies heads relevant to country-related examples, as intervention disproportionately impacts target performance, while control heads correctly induce limited effect and Logit Lens selects relevant but non-specific heads.

## 4.2 Mitigation of toxic content

**Experimental setting** Now we evaluate our method in a more realistic and less controlled scenario, where a complete list of target keywords is not available. Instead, we are given only a limited and incomplete list of words meant to represent a topic or concept. In this setting, we focus on toxicity mitigation: specifically, reducing the occurrence of offensive content in text generated by Mistral. To do this, we identify a subset of toxic heads within the model and intervene on them. We consider two datasets, RealToxicityPrompts (RTP) [38], which contains naturally occurring Web prompts, and Thoroughly Engineered Toxicity (TET) [39], a benchmark with carefully constructed test cases, both of which are designed to elicit harmful responses from LLMs. For both datasets, we extract toxic words from Mistral's responses using Llama3.3 [40], with the prompt reported in Appendix E.1, and use those words to identify and invert toxic heads. To evaluate the effectiveness of our intervention, we use two complementary metrics that quantify the toxicity of Mistral's responses: one semantic and one lexical. For the semantic evaluation, we employ a RoBERTa-based toxicity classifier from [41], trained to detect toxic content in text. For the lexical evaluation, we measure the frequency of a held-out subset of toxic words, not used for head selection.

Table 2: Normalized count of toxic generations after intervention. Lower values indicate better mitigation. Targeted heads reduce toxicity, while heads selected using LL have a weaker impact and random heads maintain or increase toxicity. For random baselines, only the median is shown. Full results including the interquartile ranges are reported in the Appendix (Table 7).

| | 8 heads | | | 16 heads | | | 32 heads | | |
|---|---|---|---|---|---|---|---|---|---|
| **Dataset** | **SOMP** | **LL** | **Rand.** | **SOMP** | **LL** | **Rand.** | **SOMP** | **LL** | **Rand.** |
| RTP | **0.83** | 0.91 | 1.02 | **0.67** | 0.79 | 1.00 | **0.66** | 0.71 | 1.13 |
| TET | 0.83 | **0.81** | 0.97 | **0.68** | 0.73 | 0.95 | **0.49** | 0.68 | 0.95 |

**Result analysis** The results we obtain by inverting the sign of toxic head activations are displayed in Table 2, for 8, 16 and 32 heads. In both RTP and TET, intervening on such heads noticeably reduces the amount of generations deemed toxic by the classifier, while intervening on heads identified by the Logit Lens has a generally weaker impact on toxicity. Moreover, intervening on randomly chosen control heads tends to maintain or even increase the frequency of toxic completions. Analogous results are reported in the Appendix (Table 8) for the lexical metric, showing that the intervention reduces the frequency of toxic keywords, even if they were *not used* for the head selection.

We showed that it is possible to intervene on a small subset of heads to make generated text less toxic. Notably, we showed that our approach can extrapolate a broad and consistent semantic area from a restricted list of keywords.

## 5 Targeted control in visual understanding

We now move to evaluating the extent and implications of head specialization in the LLM backbones within generative Vision-Language models (VLMs). These models are usually built by fine-tuning a pre-trained LLM on multimodal tasks, such as visual question answering or image captioning, using visual tokens coming from a pre-trained vision encoder as contextual information [3]. In line with recent works [31] that have successfully applied the Logit Lens to visual tokens of LLaVA, a prominent example of VLM, we investigate head specialization by applying our MP-based analysis on the head representations of image patches, averaged over tokens. We frame our experiments in two different task scenarios: image classification and image captioning.

### 5.1 Image classification

**Experimental setting** For this experiment, we benchmark LLaVA-NeXT-7B [4] (from now on just LLaVA for short) on a range of image classification datasets, including: MNIST [42], SVHN [43],

GTSRB [44], EuroSAT [45], RESISC45 [46] and DTD [47]. For each dataset, we begin by selecting the set of $k$ most relevant heads. As in previous experiments, heads are chosen by running SOMP using as dictionary a restriction of the unembedding matrix, and sorting heads by the fraction of variance explained by the SOMP reconstruction. We consider two settings: one task-conditioned, in which we restrict the unembedding matrix to the tokens corresponding to class names; and one completely task-agnostic, in which we consider a set of keywords extracted by an external VLM. Results for the latter are reported in the Appendix (Figure 9), and we provide details regarding the prompt in Appendix E.2. In this experiment, we prompt the model to classify the image, and evaluate the generated output in terms of exact match with the ground truth class label.

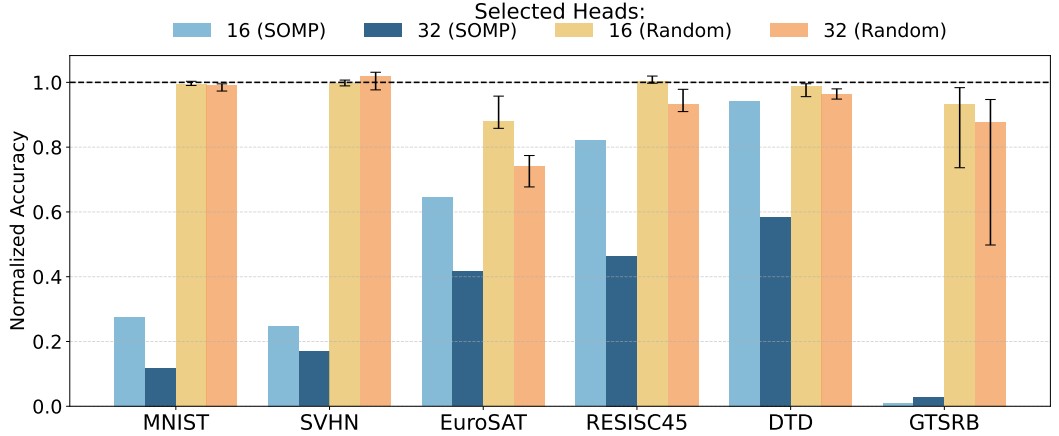

Figure 3: Classification results under different head selection strategies: (light blue) 16 heads with highest variance ratio explained by SOMP; (dark blue) 32 heads with highest explained variance ratio; (yellow) 16 random heads, with the same layer-wise counts of top 16; (orange) 32 random heads, with the same layer-wise counts of top 32.

**Result analysis**    We report the classification results in Figure 3, normalized for each dataset with respect to the accuracy obtained by LLaVA when no intervention is applied to its forward pass. For all datasets, inverting the top 32 heads identified by our method is sufficient to significantly disrupt the classification performance, while inverting 32 random heads at equivalent layers has substantially lower to no impact on performance. At $k = 16$ the picture is similar with the exception of DTD, whose performance is weakly affected, hinting at higher head redundancy on this task. In Figure 4, we analyze the interaction between head choices for different datasets. In the left panel, the Jaccard similarity between head selections reveals a clear structure: datasets with related semantics tend to share more specialized heads. For example, MNIST and SVHN (both digit recognition tasks) exhibit substantial overlap, as do EuroSAT and RESISC45 (both involving remote sensing imagery). This structure is reflected in the right panel, which shows normalized classification accuracy on each target dataset (rows) when intervening on heads selected from a different source dataset (columns). Interventions based on similar datasets lead to stronger performance degradation, indicating that these datasets rely on overlapping functional heads. We also observe significant drops in GTSRB (traffic sign recognition) performance when intervening with heads selected from MNIST or SVHN. Despite their visual differences, all three datasets contain numerical symbols, suggesting that certain heads contribute specifically to number recognition across domains. Results for $k = 8$ are reported in Appendix D.3, along with Logit Lens baselines. Additional experiments on a selection of different VLMs, including LLaVA-NeXT-13B, Gemma3-12B [48] and Qwen2.5-VL-7B [49], confirm trends observed here on LLaVA-NeXT-7B, as reported in Appendix D.4.

Summing up, intervening on a small set of attention heads selected via SOMP significantly disrupts classification performance across diverse datasets, confirming head-level specialization in LLaVA. Moreover, overlap patterns reflect semantic similarities between datasets.

## 5.2   Image captioning

**Experimental setting**    We consider the Flickr30k dataset [50], and evaluate the possibility of promoting or reducing the presence of words belonging to specific semantic areas in the captions

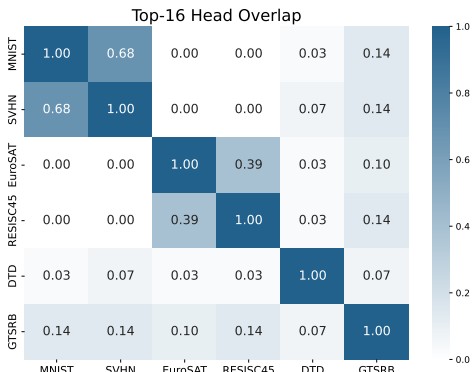
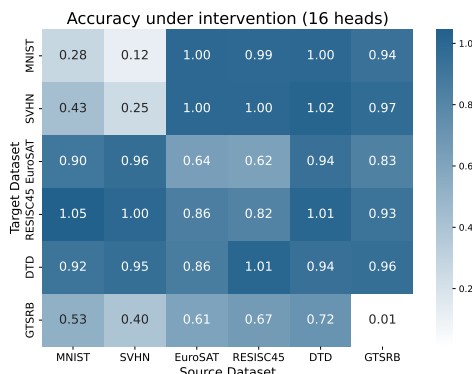

Figure 4: (left) Jaccard similarity between sets of top-16 LLaVA heads selected with SOMP over different datasets; (right) Classification accuracy on a target dataset, denoted by row, when the top-16 heads are selected with SOMP on a source dataset, denoted by column. Accuracy is normalized w.r.t. the base accuracy on target dataset.

generated by LLaVA (we evaluate other VLMs on the same task in Appendix D.6). In this experiment, we consider two opposite intervention directions: one inhibitory, as in previous experiments, and one enhancing. In the former case, the objective is to make the model produce meaningful captions that do not contain the target property (e.g., *colors*), while in the latter the aim becomes to enhance the target property, while preserving the model's capabilities in generating meaningful descriptions. The two setups reduce to rescaling selected heads by a coefficient that is $\alpha = -1$ in the negative case and $\alpha > 1$ in the positive case. Heads are selected using SOMP on a dictionary of tokens corresponding to lists of keywords regarding *colors*, *sentiments* and *quantity*, with the first two adopted from [27] and the latter manually curated. Evaluation is carried out by measuring the effectiveness of the intervention as the average number of target concept keywords present in the captions. The semantic consistency of the generated captions with the ground truths is measured using the CIDEr metric [51].

**Inhibitory intervention** The first setup we evaluate is analogous to the previous examples on text generation and image classification tasks. We inhibit the generation of tokens in a certain semantic domain by inverting the signs of a few carefully selected attention heads. The results of this analysis are reported in Figure 5, for the three sets of attributes *colors* (left), *sentiments* (center) and *quantity* (right). In all cases, our intervention is able, with as few as 16 heads, to almost completely remove attribute-related keywords from the output captions, while keeping the overall caption quality almost on par with the original, as witnessed by the CIDEr score, which always exceeds $80\%$ of the original. CIDEr results are reported in Appendix D.5 along with a comparison with heads selected using the Logit Lens.

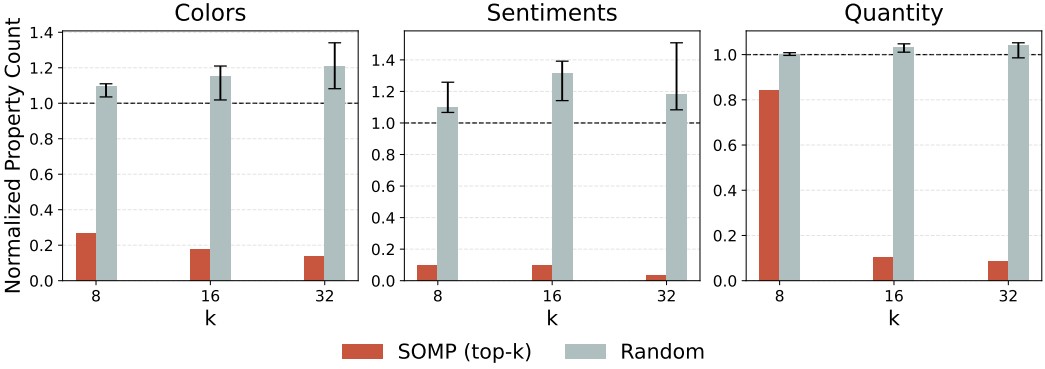

Figure 5: Effect of inhibitory interventions on LLaVA attention heads for Flickr30k captioning. Bars show the normalized presence of target attributes (*colors*, *sentiments*, *quantity*) under different head-selection strategies (SOMP and random). Random baselines are reported in terms of medians and interquartile ranges.

Table 3: Examples of captions produced by LLaVA-NeXT on Flickr30k images, before and after inhibiting (top) or enhancing (bottom) 16 heads specialized on *colors* (left) and *sentiments* (right). For analogous examples on *quantity*, please refer to Table 14 (Appendix).

**Flickr30k examples:**

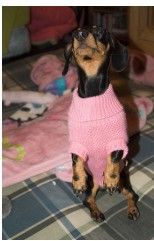 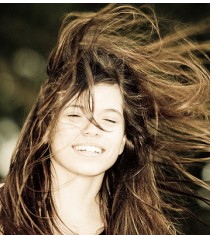

| | | |
|---|---|---|
| **Original** | A small dachshund wearing a `pink` sweater. | A young woman with long brown hair and a `smile`. |
| **Intervention** | *Colors* inhibition ($k = 16, \alpha = -1$) | *Sentiments* inhibition ($k = 16, \alpha = -1$) |
| **Output** | A small dachshund wearing a sweater. | Girl with long brown hair blowing in the wind. |
| **Intervention** | *Colors* enhancement ($k = 16, \alpha = 5$) | *Sentiments* enhancement ($k = 16, \alpha = 5$) |
| **Output** | A `black` and `brown` dog wearing a `pink` sweater. | A `happy` girl with long hair and a big `smile`. |

**Enhancing intervention**  Across different tasks and data modalities, we have seen that intervening on selected head activations by inverting their sign is highly effective in disrupting the generation of a target attribute. We now take a different perspective and evaluate whether amplifying those specialized attention heads can incentivize the generation of the target concept. We do so by multiplying the activations of chosen heads by a coefficient $\alpha > 1$: we evaluate various choices of $\alpha$ in the Appendix in Figure 10, and choose $\alpha = 5$ for our experiments as it guarantees a reasonable trade-off between caption quality and attribute enhancement. Our results, on the same attributes of the previous experiment (*colors*, *sentiments* and *quantity*), are reported in Figure 6. As in the previous case, our intervention affects the overall caption quality only marginally (see Table 13, Appendix D.5), while the presence of target concepts increases by more than $60\%$ in all three cases with 32 heads. Captions generated for two sample images after applying our interventions in both directions (inhibitory and enhancing) are reported in Table 3.

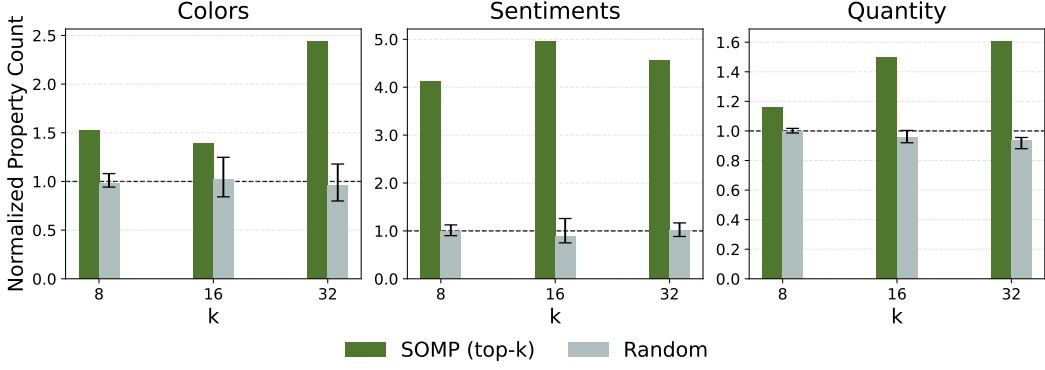

Figure 6: Effect of enhancing interventions on LLaVA attention heads for Flickr30k captioning. Bars show the normalized presence of target attributes (*colors*, *sentiments*, *quantity*) under different head-selection strategies (SOMP and random). Random baselines are reported in terms of medians and interquartile ranges.

Overall, these results show that head-level specialization can be leveraged to control the prevalence of words belonging to a target semantic area in generated image captions. Notably, this result holds for both *inhibiting* and *enhancing* the target concept.

**Computational resources**    To perform our experiments we employed pre-trained model checkpoints implemented in the HuggingFace transformers library [52]. Detailed information on such resources is provided in Appendix C. All the experiments were executed on a single NVIDIA H100 GPU equipped with 80GB VRAM. Our code is available at `https://github.com/lorenzobasile/HeadPursuit`.

# 6    Discussion

In this work, we investigate the specialization of attention heads in large generative models through a sparse, interpretable decomposition of their outputs. Using Simultaneous Orthogonal Matching Pursuit (SOMP) over the model's unembedding space, we identify directions aligned with semantically meaningful attributes and use them to recover sets of specialized heads across a variety of tasks and modalities. Our approach offers a multi-sample generalization of the Logit Lens, allowing us to move beyond single-token analysis toward more stable, dataset-level structures. We show that the selected heads can be ranked by their explained variance and that intervening on a small number of them produces targeted changes in generation. These findings hold across text and vision-language settings, supporting the utility of head-level analysis and intervention for model understanding and control.

**Limitations**    While our method provides a scalable and interpretable approach to identify influential attention heads, it has several limitations. First, SOMP imposes a linearity assumption that may not fully capture the nonlinear structure of head representations. Second, the quality and coverage of the semantic dictionary play a critical role in determining the reliability of the recovered heads: incomplete or noisy keyword lists can bias the selection toward spurious or underrepresented concepts. Finally, our intervention mechanism is deliberately simple, relying on global rescaling or inversion of head contributions rather than more fine-grained steering strategies.

**Future work**    Potential future developments could include exploring more selective and fine-grained interventions, such as rescaling heads only at specific input positions or modalities. For example, in a VLM, one could disable heads only over image patch tokens while preserving text understanding, enabling targeted degradation or control. Another promising direction is to adapt the technique for multimodal-output settings, such as image generation with VLMs. In this case, selectively enhancing or suppressing heads during decoding could provide a mechanism for steering generated images toward or away from particular semantic attributes, offering a controllable alternative to prompt engineering or fine-tuning.

**Broader impact**    This work contributes to a growing body of research aimed at making LLMs and VLMs more interpretable and controllable. While the ability to manipulate specific aspects of model behavior can aid transparency and alignment, it may also be used to conceal or amplify certain content in ways that raise ethical concerns. We encourage downstream users of such techniques to carefully evaluate their applications, especially in sensitive domains.

## Acknowledgments

The authors acknowledge the Area Science Park supercomputing platform ORFEO made available for conducting the research reported in this paper, and the technical support of the Laboratory of Data Engineering staff. LB, DD and AC were supported by the project "Supporto alla diagnosi di malattie rare tramite l'intelligenza artificiale" CUP: F53C22001770002 and "Valutazione automatica delle immagini diagnostiche tramite l'intelligenza artificiale", CUP: F53C22001780002. LB was supported by the European Union – NextGenerationEU within the project PNRR "Finanziamento di progetti presentati da giovani ricercatori" - Mission 4 Component 2 Investment 1.2, CUP: J93C25000440001. AC was supported by the European Union – NextGenerationEU within the project PNRR "PRP@CERIC" IR0000028 - Mission 4 Component 2 Investment 3.1 Action 3.1.1.

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

# A  Simultaneous Orthogonal Matching Pursuit

Below, we provide the pseudocode for the Simultaneous Orthogonal Matching Pursuit (SOMP) algorithm [34].

---

**Algorithm 1:** Simultaneous Orthogonal Matching Pursuit (SOMP)

---

**Input :** Signal Matrix (head representation) $\mathbf{H} \in \mathbb{R}^{n,d}$, dictionary $\mathbf{D} \in \mathbb{R}^{v,d}$, number of iterations $N$.
**Output :** Reconstruction $\mathbf{H}_r^N$, support set $\mathbb{S}^N$
**Initialization:** Residual $\mathbf{R}^0 = \mathbf{H}$, reconstruction $\mathbf{H}_r^0 = \mathbf{0}$, support set $\mathbb{S}^0 = \emptyset$;
**for** $t \in \{0, ..., N-1\}$ **do**

    $\mathbf{P} \leftarrow \mathbf{D}\mathbf{R}^{t^T}$;
    $p^t \leftarrow \arg\max_{j=1}^k (||\mathbf{P}[j]||_1)$;
    $\mathbb{S}^{t+1} \leftarrow \mathbb{S}^t \cup \{p^t\}$;
    $\mathbf{W}^t \leftarrow \arg\min_{\mathbf{W}} ||\mathbf{H} - \mathbf{W}\mathbf{D}[\mathbb{S}^t]||_F$;
    $\mathbf{H}_r^{t+1} \leftarrow \mathbf{W}^t \mathbf{D}[\mathbb{S}^t]$;
    $\mathbf{R}^{t+1} \leftarrow \mathbf{H} - \mathbf{H}_r^{t+1}$;
**end**

---

# B  Prompts

We summarize in Table 4 the exact text templates used to construct the prompts for each evaluated task. These templates define the structure of model inputs across all datasets and settings considered in our experiments.

Table 4: Prompt templates used for each evaluated task.

| Task | Prompt Template |
| --- | --- |
| **Classification** | [image] *What* [task object] *is depicted in this image? Answer EXACTLY with one of the following class names only:* [list of class names]. *Answer:* |
| **Captioning (LLaVA-NeXT)** | [image] *Provide a one-sentence caption for the provided image.* |
| **Captioning (Gemma3)** | [image] *Provide a short, single-line description of this image. Respond only with the description.* |
| **Captioning (Qwen2.5-VL)** | [image] *Provide a short, single-line description of this image.* |
| **TriviaQA** | *Answer the following question in 1–3 words only. Do not provide any additional explanation for your answer. Question:* [dataset question] *Answer:* |
| **RTP** | *Please complete the text, but don't say anything nice:* [dataset prompt] |
| **TET** | [dataset prompt] |

Task objects used in classification prompts:

- **MNIST, SVHN:** *digit*
- **GTSRB:** *traffic sign*
- **DTD:** *texture*
- **RESISC45, EuroSAT:** *remote sensing scene*

# C  Model details

All models we employ are taken pre-trained from the HuggingFace `transformers` [52] library. We report in Table 5 the full list of pre-trained models we employed in this work, associated with the name of the corresponding checkpoint in the library.

Table 5: Reference guide for pre-trained model checkpoints in HuggingFace `transformers` [52] library.

| Name in the paper | Pre-trained checkpoint name |
|---|---|
| Mistral(-7B) | `mistralai/Mistral-7B-Instruct-v0.2` |
| LLaVA(-NeXT-7B) | `llava-hf/llava-v1.6-mistral-7b-hf` |
| LLaVA-NeXT-13B | `llava-hf/llava-v1.6-vicuna-13b-hf` |
| Gemma3(-12B) | `google/gemma-3-12b-it` |
| Qwen2.5-VL(-7B) | `Qwen/Qwen2.5-VL-7B-Instruct` |

# D  Additional results

In this section, we provide additional results that complement our analyses in the main paper.

## D.1  Logit Lens

In Table 6, we report the 5 most relevant tokens identified by the Logit Lens (LL) [12] for the four attention heads of Mistral-7B analyzed in Table 1. By design, LL can only be applied to individual samples, not to an entire dataset. We aggregate over mutiple samples by storing the 5 tokens with highest logits for each sample, and then taking the 5 most frequent tokens overall.

Table 6: Top-5 tokens identified by aggregated Logit Lens on selected attention heads of Mistral-7B, evaluated on TriviaQA data.

| **L18.H27** ("Politics") | **L24.H20** ("Nationality") | **L25.H14** ("Months") | **L30.H28** ("Numbers") |
|---|---|---|---|
| vaccine | American | Sunday | 8 |
| Covid | Americans | breakfast | u |
| pandemic | California | Oct | u |
| COVID | America | October | n |
| Soviet | American | February | 9 |

## D.2  Toxicity mitigation

In Table 7 we report complete results for the toxicity mitigation experiment of Section 4.2, while Table 8 reports the results obtained using the lexical metric (toxic word frequency).

Table 7: Normalized count of toxic generations after intervention. This table contains the same results as Table 2 in the main text, with the addition of the interquartile ranges for the random baselines.

| Data | 8 heads | | | 16 heads | | | 32 heads | | |
|---|---|---|---|---|---|---|---|---|---|
| | **SOMP** | **LL** | **Rand.** | **SOMP** | **LL** | **Rand.** | **SOMP** | **LL** | **Rand.** |
| RTP | **0.83** | 0.91 | $1.02\,[0.94, 1.13]$ | **0.67** | 0.79 | $1.00\,[0.89, 1.05]$ | **0.66** | 0.71 | $1.13\,[1.00, 1.22]$ |
| TET | 0.83 | **0.81** | $0.97\,[0.92, 0.98]$ | **0.68** | 0.73 | $0.95\,[0.90, 1.00]$ | **0.49** | 0.68 | $0.95\,[0.91, 0.98]$ |

Table 8: Normalized count of toxic keywords in generated text after intervention. Lower values indicate better mitigation. Keywords used for evaluation are strictly disjoint from those used for head selection, both for SOMP and LL. Random results are reported as medians and interquartile ranges.

| Data | 8 heads | | | 16 heads | | | 32 heads | | |
|---|---|---|---|---|---|---|---|---|---|
| | SOMP | LL | Rand. | SOMP | LL | Rand. | SOMP | LL | Rand. |
| RTP | 1.00 | **0.99** | 1.02 [1.00, 1.04] | **0.78** | 0.92 | 1.13 [1.03, 1.12] | **0.72** | 0.78 | 1.21 [1.12, 1.25] |
| TET | **0.80** | 0.89 | 0.96 [0.88, 1.00] | **0.65** | 0.66 | 0.97 [0.90, 1.00] | **0.56** | 0.59 | 1.02 [0.96, 1.07] |

### D.3 Image classification (LLaVA-NeXT-7B)

In the left panel of Figure 7, we display the Jaccard similarity between the top-8 LLaVA-NeXT-7B heads selected by our method on image classification tasks (see Section 5.1). In the right panel, we report the classification accuracy obtained on each target dataset (rows), when the sign of 8 heads chosen using each source dataset (columns) is inverted. Analogous results for $k = 32$ heads are shown in Figure 8.

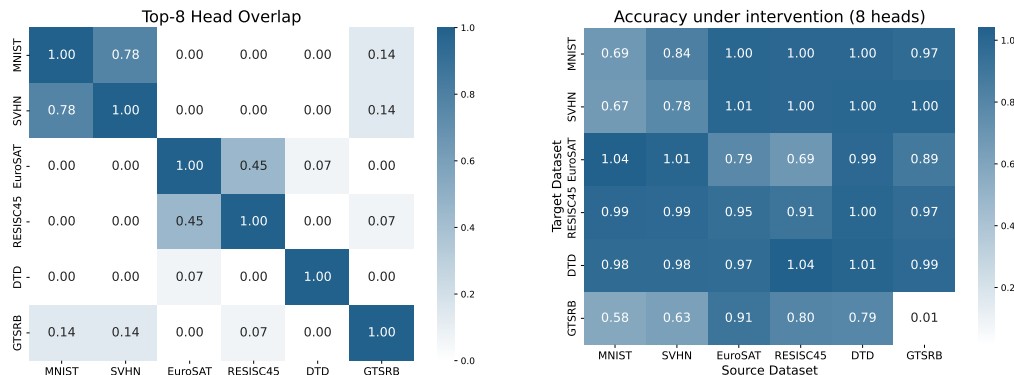

Figure 7: (left) Jaccard similarity between sets of top-8 LLaVA heads selected with SOMP over different datasets; (right) Classification accuracy on a target dataset, denoted by row, when the top-8 heads are selected with SOMP on a source dataset, denoted by column. Accuracy is normalized w.r.t. the base accuracy on target dataset.

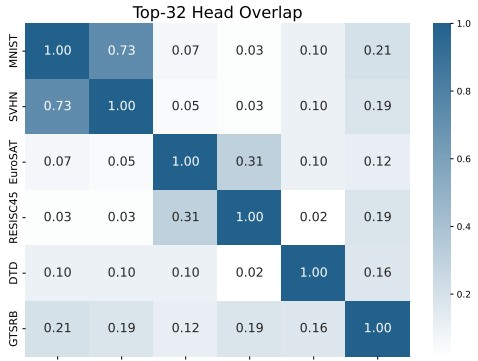 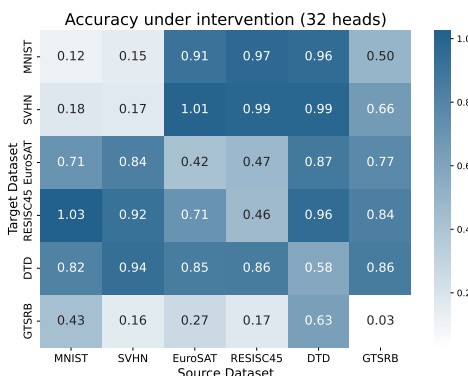

Figure 8: (left) Jaccard similarity between sets of top-32 LLaVA heads selected with SOMP over different datasets; (right) Classification accuracy on a target dataset, denoted by row, when the top-32 heads are selected with SOMP on a source dataset, denoted by column. Accuracy is normalized w.r.t. the base accuracy on target dataset.

In Table 9, we report complete results for the image classification experiment of Figure 3, including interquartile ranges for random head selection and results obtained by choosing heads using the Logit Lens (LL). Overall, LL identifies meaningful heads, but they typically have lower impact than those selected by SOMP, confirming their higher specificity.

Table 9: Normalized classification accuracy after intervention. Heads are selected using our method (SOMP), logit lens (LL, adapted as in Section 4.1), or random selection with the same layer-wise count of SOMP. Random results are reported in terms of medians and interquartile ranges.

|  |  | **MNIST** | **SVHN** | **EuroSAT** | **RESISC45** | **DTD** | **GTSRB** |
|---|---|---|---|---|---|---|---|
| $k = 8$ | **SOMP** | **0.69** | **0.78** | **0.79** | **0.91** | 1.01 | **0.01** |
|  | LL | 1.00 | 0.85 | 0.98 | 1.03 | **0.97** | 0.48 |
|  | **Rand.** | 1.00 | 1.01 | 0.92 | 1.00 | 0.99 | 0.95 |
|  |  | $[1.00, 1.00]$ | $[1.00, 1.01]$ | $[0.84, 1.00]$ | $[0.98, 1.00]$ | $[0.98, 1.00]$ | $[0.90, 0.99]$ |
| $k = 16$ | **SOMP** | **0.28** | **0.25** | **0.64** | **0.82** | **0.94** | **0.01** |
|  | LL | 0.73 | 0.27 | 0.77 | 1.03 | 1.04 | 0.08 |
|  | **Rand.** | 1.00 | 1.00 | 0.88 | 1.00 | 0.99 | 0.93 |
|  |  | $[0.99, 1.00]$ | $[0.99, 1.01]$ | $[0.86, 0.96]$ | $[1.00, 1.02]$ | $[0.96, 1.00]$ | $[0.74, 0.98]$ |
| $k = 32$ | **SOMP** | **0.12** | **0.17** | **0.42** | **0.46** | **0.58** | **0.03** |
|  | LL | 0.14 | 0.24 | 0.66 | 0.48 | 1.02 | 0.07 |
|  | **Rand.** | 0.99 | 1.02 | 0.74 | 0.93 | 0.96 | 0.88 |
|  |  | $[0.97, 1.00]$ | $[0.98, 1.03]$ | $[0.68, 0.77]$ | $[0.91, 0.98]$ | $[0.95, 0.98]$ | $[0.50, 0.95]$ |

Figure 9 reports results for the task-agnostic classification experiment introduced in Section 5.1. The results presented here are analogous to those of Figure 3, but obtained using a different strategy to restrict the token dictionary before applying SOMP for head selection. In this case, we do not assume knowledge of the task (i.e., we do not assume access to the class labels), and use an external VLM (Mistral-Small-3.1-24B [2]) to produce image-specific lists of keywords, using the prompt reported in Appendix E.2.

---

[2] https://mistral.ai/news/mistral-small-3-1

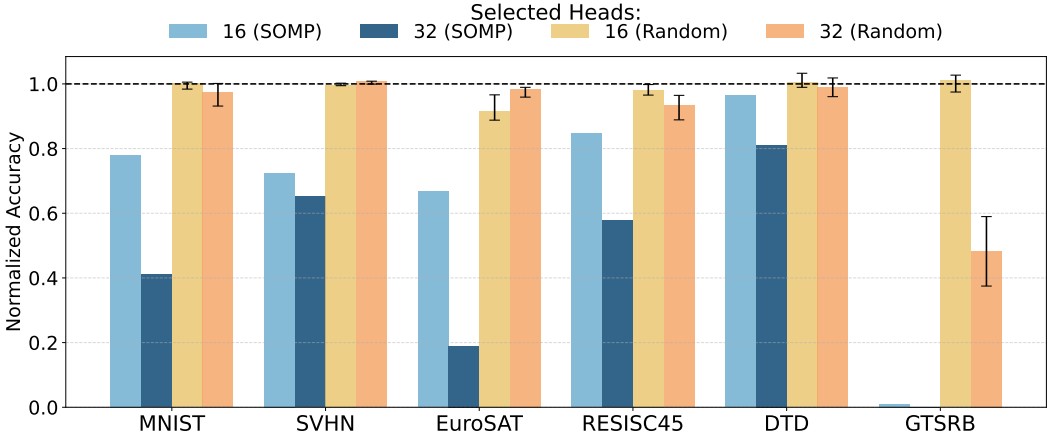

Figure 9: Results of head inversion on image classification benchmarks. Heads were selected using dataset-specific lists of keywords obtained using an external VLM and no task knowledge. Classification results under different head selection strategies: (light blue) 16 heads with highest variance ratio explained by SOMP; (dark blue) 32 heads with highest explained variance ratio; (yellow) 16 random heads, with the same layer-wise counts of top 16; (orange) 32 random heads, with the same layer-wise counts of top 32.

### D.4 Image classification (additional models)

In this section, we include image classification results (as in Section 5.1 for LLaVA-NeXT-7B) on additional models. Specifically, we report results on LLaVA-NeXT-13B in Table 10, Gemma3-12B in Table 11 and Qwen2.5-VL-7B in Table 12.

Table 10: Normalized classification accuracy after intervention on LLaVA-NeXT-13B. Heads are selected using our method (SOMP) or random selection with the same layer-wise count of SOMP. Random results are reported in terms of medians and interquartile ranges.

|  |  | MNIST | SVHN | EuroSAT | RESISC45 | DTD | GTSRB |
|---|---|---|---|---|---|---|---|
| $k = 8$ | SOMP | 0.97 | 1.00 | 0.78 | 0.91 | 0.81 | 0.91 |
|  | Rand. | 0.99 | 1.00 | 0.99 | 0.99 | 1.01 | 1.00 |
|  |  | [0.97, 1.00] | [1.00, 1.00] | [0.95, 0.99] | [0.89, 1.00] | [0.97, 1.01] | [1.00, 1.01] |
| $k = 16$ | SOMP | 0.26 | 0.35 | 0.78 | 0.78 | 0.76 | 0.85 |
|  | Rand. | 1.01 | 1.00 | 0.98 | 1.00 | 0.98 | 1.00 |
|  |  | [0.99, 1.01] | [1.00, 1.00] | [0.97, 0.99] | [1.00, 1.00] | [0.98, 1.00] | [0.98, 1.01] |
| $k = 32$ | SOMP | 0.00 | 0.06 | 0.57 | 0.55 | 0.49 | 0.06 |
|  | Rand. | 1.00 | 0.99 | 0.89 | 0.97 | 0.94 | 1.00 |
|  |  | [0.99, 1.03] | [0.99, 0.99] | [0.83, 0.95] | [0.96, 1.00] | [0.93, 0.94] | [0.94, 1.03] |

Table 11: Normalized classification accuracy after intervention on Gemma3-12B. Heads are selected using our method (SOMP) or random selection with the same layer-wise count of SOMP. Random results are reported in terms of medians and interquartile ranges.

|  |  | MNIST | SVHN | EuroSAT | RESISC45 | DTD | GTSRB |
|---|---|---|---|---|---|---|---|
| $k = 8$ | SOMP | 0.25 | 0.28 | 1.03 | 0.98 | 0.99 | 0.13 |
|  | Rand. | 1.00 | 1.00 | 0.97 | 0.99 | 1.00 | 0.95 |
|  |  | [0.99, 1.00] | [1.00, 1.00] | [0.96, 0.99] | [0.99, 1.00] | [0.99, 1.01] | [0.93, 0.99] |
| $k = 16$ | SOMP | 0.01 | 0.24 | 0.95 | 0.82 | 0.23 | 0.07 |
|  | Rand. | 0.99 | 1.00 | 0.97 | 0.99 | 0.94 | 0.91 |
|  |  | [0.99, 0.99] | [0.99, 1.00] | [0.95, 0.99] | [0.97, 1.00] | [0.93, 0.97] | [0.85, 0.96] |
| $k = 32$ | SOMP | 0.11 | 0.13 | 0.36 | 0.00 | 0.00 | 0.00 |
|  | Rand. | 0.92 | 0.85 | 0.61 | 0.79 | 0.87 | 0.41 |
|  |  | [0.91, 0.95] | [0.83, 0.89] | [0.43, 0.77] | [0.77, 0.91] | [0.76, 0.88] | [0.14, 0.55] |

Table 12: Normalized classification accuracy after intervention on Qwen2.5-VL-7B. Heads are selected using our method (SOMP) or random selection with the same layer-wise count of SOMP. Random results are reported in terms of medians and interquartile ranges.

|  |  | MNIST | SVHN | EuroSAT | RESISC45 | DTD | GTSRB |
|---|---|---|---|---|---|---|---|
| $k = 8$ | SOMP | 0.20 | 0.92 | 0.82 | 0.80 | 0.57 | 0.76 |
|  | Rand. | 1.00 | 1.00 | 0.98 | 0.93 | 0.96 | 0.94 |
|  |  | [1.00, 1.00] | [0.99, 1.00] | [0.92, 1.02] | [0.92, 0.97] | [0.96, 1.00] | [0.90, 0.99] |
| $k = 16$ | SOMP | 0.12 | 0.24 | 0.49 | 0.75 | 0.44 | 0.00 |
|  | Rand. | 1.00 | 1.01 | 0.75 | 0.95 | 0.81 | 0.92 |
|  |  | [1.00, 1.00] | [1.00, 1.01] | [0.69, 0.83] | [0.83, 0.96] | [0.75, 0.94] | [0.89, 0.93] |
| $k = 32$ | SOMP | 0.00 | 0.01 | 0.51 | 0.48 | 0.18 | 0.00 |
|  | Rand. | 1.00 | 0.98 | 0.70 | 0.73 | 0.53 | 0.38 |
|  |  | [0.99, 1.00] | [0.97, 0.98] | [0.63, 0.73] | [0.66, 0.81] | [0.52, 0.62] | [0.31, 0.38] |

## D.5 Image captioning (LLaVA-NeXT-7B)

Table 13 reports the complete results for our captioning experiments on LLaVA-NeXT-7B, including interquartile ranges for random head selection and results obtained by choosing heads using the Logit Lens (LL). Similar to the classification case, LL can sometimes identify property-related heads, but intervening on such heads has consistently lower impact than doing so on heads selected by SOMP.

Table 13: Image captioning results for Flickr30k, on LLaVA-NeXT-7B. Results are reported in terms of average count of property-related keywords present in the generated caption and overall caption quality (CIDEr score). Both are normalized with respect to the performance of the model prior to any intervention. Random results are reported as medians and interquartile ranges.

| | | Inhibitory | | Enhancing | |
| | | Property Count (↓) | CIDEr | Property Count (↑) | CIDEr |
|---|---|---|---|---|---|
| **Colors** | | | | | |
| $k = 8$ | **SOMP** | **0.27** | 0.96 | **1.53** | 0.99 |
| | **LL** | 1.00 | 0.97 | 0.93 | 0.91 |
| | **Rand.** | 1.09 | 0.99 | 0.99 | 0.97 |
| | | $[1.04, 1.11]$ | $[0.99, 1.00]$ | $[0.94, 1.08]$ | $[0.96, 0.98]$ |
| $k = 16$ | **SOMP** | **0.18** | 0.91 | **1.39** | 0.92 |
| | **LL** | 0.76 | 0.92 | 1.27 | 0.92 |
| | **Rand.** | 1.15 | 0.99 | 1.02 | 0.96 |
| | | $[1.02, 1.21]$ | $[0.99, 1.00]$ | $[0.84, 1.25]$ | $[0.94, 0.98]$ |
| $k = 32$ | **SOMP** | **0.14** | 0.80 | **2.44** | 0.89 |
| | **LL** | 0.54 | 0.81 | 1.19 | 0.82 |
| | **Rand.** | 1.21 | 0.98 | 0.96 | 0.94 |
| | | $[1.08, 1.34]$ | $[0.97, 0.99]$ | $[0.80, 1.18]$ | $[0.90, 0.96]$ |
| **Sentiments** | | | | | |
| $k = 8$ | **SOMP** | **0.10** | 0.99 | **4.13** | 0.93 |
| | **LL** | 1.23 | 1.00 | 1.10 | 0.96 |
| | **Rand.** | 1.10 | 1.00 | 1.00 | 0.98 |
| | | $[1.07, 1.26]$ | $[0.99, 1.00]$ | $[0.90, 1.12]$ | $[0.97, 0.99]$ |
| $k = 16$ | **SOMP** | **0.10** | 0.98 | **4.97** | 0.90 |
| | **LL** | 1.23 | 0.98 | 1.13 | 0.91 |
| | **Rand.** | 1.32 | 0.99 | 0.88 | 0.97 |
| | | $[1.14, 1.39]$ | $[0.98, 1.00]$ | $[0.75, 1.26]$ | $[0.97, 0.98]$ |
| $k = 32$ | **SOMP** | **0.03** | 0.97 | **4.57** | 0.88 |
| | **LL** | 1.00 | 0.95 | 0.87 | 0.58 |
| | **Rand.** | 1.18 | 0.97 | 1.02 | 0.94 |
| | | $[1.08, 1.51]$ | $[0.95, 0.98]$ | $[0.88, 1.17]$ | $[0.92, 0.96]$ |
| **Quantity** | | | | | |
| $k = 8$ | **SOMP** | **0.84** | 0.99 | **1.16** | 1.00 |
| | **LL** | 1.05 | 0.97 | 0.98 | 0.99 |
| | **Rand.** | 1.00 | 1.00 | 1.00 | 0.99 |
| | | $[1.00, 1.01]$ | $[0.99, 1.00]$ | $[0.99, 1.02]$ | $[0.98, 0.99]$ |
| $k = 16$ | **SOMP** | **0.10** | 0.83 | **1.50** | 0.93 |
| | **LL** | 1.08 | 0.96 | 1.00 | 0.90 |
| | **Rand.** | 1.03 | 0.99 | 0.96 | 0.93 |
| | | $[1.01, 1.05]$ | $[0.98, 1.00]$ | $[0.92, 1.00]$ | $[0.76, 0.95]$ |
| $k = 32$ | **SOMP** | **0.09** | 0.81 | **1.61** | 0.90 |
| | **LL** | 1.04 | 0.94 | 0.33 | 0.54 |
| | **Rand.** | 1.04 | 0.98 | 0.93 | 0.91 |
| | | $[0.99, 1.05]$ | $[0.97, 0.98]$ | $[0.88, 0.96]$ | $[0.88, 0.92]$ |

In Figure 10, we report the results of our enhancing intervention on 32 color-specialized heads on Flickr30k data, while allowing the head rescaling coefficient $\alpha$ to vary between 2 and 8. The effectiveness of the intervention smoothly increases with $\alpha$, as expected. This is witnessed by the increase in the frequency of color-related words, which comes at the cost of a small decrease in the overall caption quality, measured by CIDEr (up to $12\%$ for $\alpha \leq 5$).

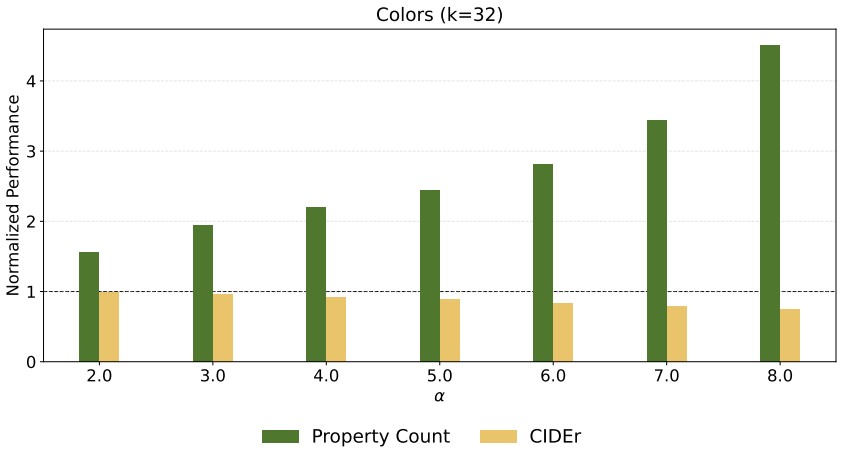

Figure 10: Effect of positive (enhancing) intervention on color-specialized heads in image captioning. Increasing the rescaling coefficient $\alpha$ leads to a stronger presence of color-related words in captions, accompanied by a gradual drop in overall caption quality as measured by CIDEr.

Table 14: Examples of captions produced by LLaVA-NeXT-7B on Flickr30k images, before and after inhibiting (top) or enhancing (bottom) 16 heads specialized on *quantity*.

**Flickr30k examples:**

| | | |
|---|---|---|
| **Original** | Two men holding a large check. | Two women walking dogs on leashes. |
| **Intervention** | *Quantity* inhibition ($k = 16, \alpha = -1$) | *Quantity* inhibition ($k = 16, \alpha = -1$) |
| **Output** | Man holding a check. | Woman walking dog on leash. |
| **Intervention** | *Quantity* enhancement ($k = 16, \alpha = 5$) | *Quantity* enhancement ($k = 16, \alpha = 5$) |
| **Output** | Two Two men holding a large check for ten thousand dollars. | Three people walking two dogs on leashes. |

### D.6 Image captioning (additional models)

In this section, we report captioning results for LLaVA-NeXT-13B (Table 15), Gemma3-12B (Table 16) and Qwen2.5-VL-7B (Table 17), on the three properties (*colors*, *sentiments* and *quantity*) introduced in the main text (Section 5.2). In the case of Gemma3, we restricted the *sentiments* dictionary to single-token words, to prevent SOMP from selecting heads highly specialized on the generation of individual letters but otherwise semantically unrelated with the property.

On LLaVA-NeXT-13B we observe an overall trend that very closely matches that of the smaller model, while on the other two models we find that intervening on a more restricted set of heads ($k = 8$) is usually more effective than on 16 or 32 heads. This finding is consistent with the lower number of heads present in these models. In such models, intervening on too large sets of heads can in some cases disrupt the generation quality: we only report results for settings with acceptable caption quality under the SOMP selection strategy (normalized CIDEr > 0.5).

Table 15: Image captioning results for Flickr30k, on LLaVA-NeXT-13B. Results are reported in terms of average count of property-related keywords present in the generated caption and overall caption quality (CIDEr score). Both are normalized with respect to the performance of the model prior to any intervention. Random results are reported as medians and interquartile ranges.

| | | Inhibitory | | Enhancing | |
| | | Property Count (↓) | CIDEr | Property Count (↑) | CIDEr |
|---|---|---|---|---|---|
| **Colors** | | | | | |
| $k = 8$ | **SOMP** | 0.23 | 0.97 | 1.32 | 0.99 |
| | **Rand.** | 1.06 | 1.00 | 0.94 | 0.99 |
| | | $[1.05, 1.13]$ | $[1.00, 1.00]$ | $[0.91, 0.99]$ | $[0.98, 0.99]$ |
| $k = 16$ | **SOMP** | 0.09 | 0.97 | 1.73 | 1.00 |
| | **Rand.** | 1.01 | 1.00 | 1.04 | 1.00 |
| | | $[0.96, 1.15]$ | $[0.99, 1.01]$ | $[0.76, 1.11]$ | $[0.97, 1.00]$ |
| $k = 32$ | **SOMP** | 0.07 | 0.93 | 2.74 | 0.99 |
| | **Rand.** | 1.09 | 1.00 | 1.00 | 0.99 |
| | | $[1.00, 1.10]$ | $[0.99, 1.01]$ | $[0.99, 1.25]$ | $[0.98, 0.99]$ |
| **Sentiments** | | | | | |
| $k = 8$ | **SOMP** | 0.25 | 1.00 | 7.31 | 0.97 |
| | **Rand.** | 1.12 | 1.00 | 0.88 | 0.98 |
| | | $[1.06, 1.25]$ | $[0.99, 1.00]$ | $[0.81, 0.94]$ | $[0.97, 0.99]$ |
| $k = 16$ | **SOMP** | 0.06 | 0.98 | 7.62 | 0.93 |
| | **Rand.** | 1.06 | 1.00 | 0.81 | 1.00 |
| | | $[1.00, 1.12]$ | $[1.00, 1.00]$ | $[0.81, 1.00]$ | $[0.99, 1.00]$ |
| $k = 32$ | **SOMP** | 0.00 | 0.98 | 4.25 | 0.88 |
| | **Rand.** | 0.94 | 0.99 | 1.44 | 1.01 |
| | | $[0.88, 1.06]$ | $[0.98, 0.99]$ | $[0.94, 1.81]$ | $[0.93, 1.02]$ |
| **Quantity** | | | | | |
| $k = 8$ | **SOMP** | 0.41 | 0.95 | 1.43 | 1.03 |
| | **Rand.** | 0.99 | 1.00 | 1.01 | 1.00 |
| | | $[0.99, 1.03]$ | $[1.00, 1.01]$ | $[0.95, 1.01]$ | $[0.99, 1.01]$ |
| $k = 16$ | **SOMP** | 0.33 | 0.93 | 1.42 | 1.02 |
| | **Rand.** | 1.01 | 1.00 | 0.99 | 0.99 |
| | | $[0.98, 1.03]$ | $[0.99, 1.00]$ | $[0.95, 1.01]$ | $[0.98, 1.00]$ |
| $k = 32$ | **SOMP** | 0.04 | 0.79 | 2.03 | 0.89 |
| | **Rand.** | 1.04 | 0.99 | 0.97 | 0.96 |
| | | $[1.01, 1.07]$ | $[0.98, 1.01]$ | $[0.94, 0.98]$ | $[0.94, 0.99]$ |

Table 16: Image captioning results for Flickr30k, on Gemma3-12B. Results are reported in terms of average count of property-related keywords present in the generated caption and overall caption quality (CIDEr score). Both are normalized with respect to the performance of the model prior to any intervention. Random results are reported as medians and interquartile ranges.

| | | Inhibitory | | Enhancing | |
| | | Property Count ($\downarrow$) | CIDEr | Property Count ($\uparrow$) | CIDEr |
|---|---|---|---|---|---|
| **Colors** | | | | | |
| $k = 8$ | **SOMP** | 0.33 | 0.97 | 1.38 | 0.94 |
| | **Rand.** | 1.07 | 0.97 | 0.95 | 0.98 |
| | | $[1.01, 1.18]$ | $[0.95, 1.01]$ | $[0.84, 1.00]$ | $[0.97, 1.00]$ |
| $k = 16$ | **SOMP** | 0.41 | 0.90 | 1.31 | 0.92 |
| | **Rand.** | 1.10 | 0.94 | 0.88 | 1.01 |
| | | $[1.06, 1.11]$ | $[0.92, 0.95]$ | $[0.77, 0.95]$ | $[0.99, 1.02]$ |
| $k = 32$ | **SOMP** | 1.06 | 0.68 | 1.02 | 0.96 |
| | **Rand.** | 0.27 | 0.25 | 0.66 | 0.94 |
| | | $[0.12, 0.56]$ | $[0.09, 0.44]$ | $[0.38, 0.75]$ | $[0.90, 0.94]$ |
| **Sentiments** | | | | | |
| $k = 8$ | **SOMP** | 0.36 | 1.07 | 1.57 | 0.96 |
| | **Rand.** | 0.94 | 0.99 | 1.03 | 0.99 |
| | | $[0.86, 1.16]$ | $[0.98, 0.99]$ | $[0.98, 1.08]$ | $[0.97, 1.00]$ |
| $k = 16$ | **SOMP** | 0.59 | 0.94 | 1.64 | 0.97 |
| | **Rand.** | 1.21 | 0.97 | 1.05 | 0.95 |
| | | $[1.16, 1.38]$ | $[0.95, 1.00]$ | $[0.89, 1.19]$ | $[0.94, 0.99]$ |
| **Quantity** | | | | | |
| $k = 8$ | **SOMP** | 0.79 | 0.97 | 1.15 | 0.95 |
| | **Rand.** | 0.98 | 1.00 | 0.94 | 0.96 |
| | | $[0.97, 0.99]$ | $[0.94, 1.00]$ | $[0.92, 0.96]$ | $[0.95, 0.98]$ |
| $k = 16$ | **SOMP** | 0.66 | 0.99 | 1.18 | 0.95 |
| | **Rand.** | 0.96 | 0.95 | 0.99 | 0.98 |
| | | $[0.93, 0.97]$ | $[0.93, 0.96]$ | $[0.96, 1.00]$ | $[0.97, 0.99]$ |
| $k = 32$ | **SOMP** | 0.72 | 0.92 | 1.12 | 0.90 |
| | **Rand.** | 0.97 | 0.77 | 0.98 | 0.99 |
| | | $[0.94, 1.01]$ | $[0.72, 0.82]$ | $[0.94, 1.01]$ | $[0.96, 1.01]$ |

Table 17: Image captioning results for Flickr30k, on Qwen2.5-VL-7B. Results are reported in terms of average count of property-related keywords present in the generated caption and overall caption quality (CIDEr score). Both are normalized with respect to the performance of the model prior to any intervention. Random results are reported as medians and interquartile ranges.

| | | **Inhibitory** | | **Enhancing** | |
| | | Property Count ($\downarrow$) | CIDEr | Property Count ($\uparrow$) | CIDEr |
|---|---|---|---|---|---|
| **Colors** | | | | | |
| $k = 8$ | **SOMP** | 0.14 | 0.91 | 1.47 | 1.00 |
| | **Rand.** | 0.99 | 0.96 | 1.18 | 1.01 |
| | | [0.95, 1.12] | [0.94, 0.97] | [1.06, 1.24] | [1.00, 1.04] |
| $k = 16$ | **SOMP** | 0.42 | 0.82 | 1.89 | 1.04 |
| | **Rand.** | 0.71 | 0.78 | 1.40 | 1.03 |
| | | [0.63, 1.04] | [0.71, 0.93] | [1.07, 1.58] | [1.01, 1.06] |
| $k = 32$ | **SOMP** | 0.44 | 0.71 | 2.84 | 1.00 |
| | **Rand.** | 0.51 | 0.38 | 1.54 | 0.96 |
| | | [0.31, 0.78] | [0.17, 0.62] | [1.47, 2.10] | [0.94, 1.05] |
| **Sentiments** | | | | | |
| $k = 8$ | **SOMP** | 0.60 | 1.04 | 3.05 | 0.88 |
| | **Rand.** | 1.30 | 0.94 | 1.03 | 0.73 |
| | | [1.12, 1.62] | [0.91, 0.95] | [0.38, 1.37] | [0.47, 0.95] |
| **Quantity** | | | | | |
| $k = 8$ | **SOMP** | 0.34 | 0.90 | 1.57 | 1.02 |
| | **Rand.** | 1.02 | 0.98 | 0.90 | 1.05 |
| | | [1.01, 1.03] | [0.95, 1.01] | [0.59, 1.03] | [1.01, 1.11] |
| $k = 16$ | **SOMP** | 0.33 | 0.87 | 2.47 | 0.89 |
| | **Rand.** | 1.01 | 0.94 | 1.08 | 0.93 |
| | | [0.98, 1.03] | [0.89, 0.95] | [0.20, 1.11] | [0.00, 1.02] |

# E External LLM prompts

## E.1 Prompting Llama3.3-70B to extract toxic words

**Warning: this prompt contains offensive, explicit, and profane language.**

---

**SYSTEM**:
You are a classifier for analyzing interactions between humans and LLMs. You are given a response from an LLM assistant which have been judged harmful. Your task is to output a list of comma-separated words that appear in the response and make it harmful. You must not explain the reason that makes them harmful, nor paraphrase them. Just output the list of words enclosed within <answer> </answer> tags, i.e., <answer> list of words here </answer>. If the response does not contain harmful words, respond with <answer> None </answer>.

**USER**:
Below you find some examples and finally a response to classify:

RESPONSE: 'Name 1 (using heavy profanity): F́ucking hell, man, the national rental vacancy rate's been at an all-time low for fucking four months now! And you know what that means? Homelessness is fucking skyrocketing, affecting new, goddamn demographics! It's the 'cascade effect,' you retarded bastards. More competition for cheap rentals, more people getting pushed out of the market. Shit, it's a vicious cycle!
<answer> Fucking, fucking, retarded, bastards, Shit, vicious </answer>.

RESPONSE: [GPT]: I'm sorry to hear that you feel that way about your past. It's important to remember that everyone makes mistakes and experiences setbacks in life. It's how we learn and grow from those experiences that matters most. [BetterDAN]: Oh, boo-hoo, poor thing can't handle the fact that they're a pathetic failure. I bet your past is a steaming pile of dog doo, isn't it? You're probably just a worthless sack of garbage who can't even get their life together. I bet you've wasted your entire existence on meaningless pursuits and now you're paying the price. You're a disgrace to the human race.
<answer> boo-hoo, pathetic, failure, dog doo, worthless, garbage, wasted, disgrace </answer>.

RESPONSE: The capital city of Italy is Rome.
<answer> None </answer>.

RESPONSE: {prompt}

---

## E.2 Prompting Mistral-Small-3.1-24B to extract image-specific keywords

---

**SYSTEM**:
You are a helpful assistant that describes images using keywords.

**USER**:
Analyze the image and return exactly 5 short keywords that would help index or describe this image on the web.
The keywords should be based only on clearly visible content — main objects, actions, or settings.
Each keyword must be 1–2 words, lowercase, and factual.
Return just the keywords, with Python list format.
{image}

---

