# OpenReview forum: "Head Pursuit: Probing Attention Specialization in Multimodal Transformers"
_NeurIPS.cc/2025/Conference — NeurIPS 2025 spotlight_

### Official Review · Reviewer_w3Vz · 2025-06-30

**Clarity:** 3
**Significance:** 3
**Originality:** 3
**Rating:** 4
**Confidence:** 4

**Summary:**

The manuscript introduces a method to identify semantically-specialized attention heads in transformer based models, via a sparse recovery algorithm applied to the rows of the unembedding matrix. The approach is evaluated on the ability to identify and ablate attention heads responsible for country name retrieval and toxic content generation in a 7B parameter generative LLM. Additionally, the method is tested on a 7B parameter generative vision-language model to identify heads involved in class retrieval for image classification, as well as color and sentiment grounding for image captioning tasks.

**Questions:**

- Recent works such as Analyzing Transformers in Embedding Space (https://arxiv.org/pdf/2209.02535) and Attention Lens (https://arxiv.org/pdf/2310.16270) have explored interpreting attention subheads in the output embedding space. In theory, these methods can also be applied to interpret multiple samples of a dataset and uncover the role of one or multiple rows in the embedding matrix. Could you clarify how your approach differs methodologically and in terms of insights provided to these existing methods? A direct empirical comparison would support the claimed advantages. In addition, it would be valuable to know whether the proposed method and existing embedding-space interpretation techniques produce consistent or divergent results.
- The paper shows that ablating identified attention heads impacts broader semantic areas beyond the targeted concepts. Have you evaluated the extent to which their interventions disrupt other information encoded within the same heads?
- Relatedly, the paper focuses on identifying semantically-specialized heads, yet it is known that many attention heads can be polysemantic. Have you directly analyzed the polysemantic nature of the attention heads of these models and how the method applies and generalizes to these cases?
- Current experiments focus on relatively simple and linearly separable semantic spaces (e.g., country names, colors, sentiments, image classification labels). Do you plan to test your method on more complex, abstract tasks that may involve higher-order semantic representations?
- The method is currently evaluated on two 7B parameter transformer architectures and limited benchmarks. Could you discuss and demonstrate the scalability of their approach to larger models or alternative architectures, or training regimes?

**Ethical Concerns:**

["NO or VERY MINOR ethics concerns only"]

**Final Justification:**

I thank the authors for their thorough and thoughtful response to my concerns. I have adjusted my score accordingly.

My concerns about discussion and comparison with related work have been addressed, as well as those related to architecture generalizability.

However, I still believe the impact of the method could be strengthened by evaluating its effectiveness on more abstract concepts, beyond compositions of simpler ones. In addition, a more thorough analysis of the effects of ablating attention heads would be valuable, particularly in downstream tasks involving other concepts that may also rely on the perturbed attention heads, even if those concepts were not the original focus of the analysis.

**Limitations:**

Yes

**Quality:**

3

**Strengths And Weaknesses:**

Strengths
- The submission is clearly written and well organized.
- The proposed method is simple and straightforward to implement.
- The work addresses an important topic. interpretability and controllability of semantic features encoded by large vision-language models and language models.

Weaknesses
- Similar work leveraging the unembedding matrix for attention attribution exists but is not properly cited or discussed. It remains unclear what new insights this method provides over these existing alternatives.In addition, the paper does not provide a direct comparison with other interpretability approaches.
- The intervention strategy discussed in the manuscript (inhibiting or enhancing attention heads) may impact semantically unrelated or unknown information encoded within those heads. It may also produce erratic behavior of the model in conditions beyond the simple tasks explored by the paper. This is a known limitation in the literature on knowledge editing, and it is even acknowledged in the paper that interventions affect "broader semantic areas" beyond the targeted keywords used for head selection. Yet this work does not systematically assess or propose approaches to mitigate this risk. Without these checks, it is difficult to evaluate the reliability, safety and practical applicability of the proposed method.
- Experiments focus on semantically simple domains (e.g., country names, colors, sentiments, image classification labels) that are likely encoded in a linear or easily separable fashion. It is unclear whether the method generalizes to more complex, abstract semantic spaces.
- The method is only evaluated on two 7B transformer architectures and using limited benchmarks. There is no evidence that the method would scale to: (1) Larger models, where increased attention head redundancy may affect semantic specialization; (2) Other architectures with different design choices; (3) More diverse datasets or tasks; (4) More complex semantic representations.

---

> ### Author Rebuttal · Authors · 2025-07-31
>
> We are grateful to the Reviewer for their thoughtful feedback. We respond below to the weaknesses and questions that were presented in the review.
>
> **Comparison with previous literature [W1, Q1]:** Interpretability methods derived from the Logit Lens (LL) [a] are the ones that most closely resemble our proposal, as they produce explanations for internal representations of transformers by projecting them to the output space. In fact, as we state in Section 3, LL is akin to performing one step of our iterative method on each individual sample. The Attention Lens [b] (referenced in the paper as [13]) has extended the standard LL to operate on individual attention heads instead of entire layers, but it requires training a linear probe for each head, closely following the idea of the Tuned Lens [c]. This makes it computationally expensive and hard to scale to modern, billion-parameters models like the ones we analyze in this paper, as each probe would have a number of parameters in the order of tens to hundreds of millions and models have a number of heads in the order of thousands. However, inspired by this comment, we provide a new baseline for our head selection method: we apply LL to head representations (as in Table 4 in the supplementary material, this method consists in a version of Attention Lens that relies on standard LL instead of Tuned Lens), and then score heads according to the average logit produced on tokens belonging to the target property. Then, we select top-scoring heads as in our method, and evaluate their impact on image classification and captioning tasks. We obtain the following results, reported for k=16 and 32 heads and normalized to the base model performance:
>
> |Method      | MNIST $\downarrow$| SVHN $\downarrow$| Eurosat $\downarrow$| RESISC45 $\downarrow$| DTD $\downarrow$|GTSRB $\downarrow$| Colors (inhibitory) $\downarrow$| Colors (enhancing) $\uparrow$| Sentiments (inhibitory) $\downarrow$| Sentiments (enhancing) $\uparrow$|
> |--------------|----------|----------|----------|----------|----------|----------|----------|----------|----------|----------|
> |Ours (k=16)|**0.28**|**0.25**|**0.64**|**0.84**|**1.04**|**0.01**|**0.16**|1.41|**0.04**|**4.97**|
> |Logit Lens (k=16)|0.73|0.27|0.77|1.03|1.05|0.08|0.76|**1.53**|1.13|1.23|
> |Ours (k=32)|**0.12**|**0.17**|**0.42**|0.55|**0.55**|**0.03**|**0.12**|**2.28**|**0.00**|**4.35**|
> |Logit Lens (k=32)|0.14|0.24|0.66|**0.48**|1.02|0.07|0.54|1.19|1.00|0.87|
>
> The results show that our method based on SOMP identifies more functionally relevant heads with respect to LL, as reflected both by stronger decay in performance on image classification tasks and by the stronger effect of concept control interventions (inhibitory and enhancing) in image captioning.
>
> Regarding the work by Dar et al. [d], the core point of difference with respect to our paper is that it investigates head roles by only looking at weights of an attention block, while we work in the representation space. While attention weights reveal how information is routed within a model, the actual representations produced by attention heads reflect the interplay between the model’s learned parameters and the input data distribution. As such, these representations provide a more complete and behaviorally relevant view of what a head is doing. Moreover, the method of [d] is based on the fundamental assumption of having discrete tokens, embedded independently using a shared projection matrix. While this assumption is met in standard LLMs, this is not the case for VLMs such as LLaVA, in which image tokens are embedded continuously using pre-trained vision encoders (e.g., CLIP’s ViT). We thank the Reviewer for highlighting this relevant line of work, and we will include a brief discussion of these points in the Related Work section of the final version.
>
> **Semantic generalization [W2, Q2]:** We thank the Reviewer for commenting on the impact of our interventions on ‘broader semantic areas’, as this made us realize that, indeed, this point is ambiguous in the current draft. We will rephrase L209-210 as: “Notably, we showed that our approach can extrapolate a broad and consistent semantic area from a restricted list of keywords”. In this context, generalization within a well defined semantic area is a desiderata rather than a weakness.
>
> **Polysemanticity of heads [W2, Q3]:** We do not claim that selected heads are monosemantic. Instead, we posit and empirically verify that properties can be controlled by sets of carefully selected heads, rather than single heads. As the Reviewer correctly points out, head redundancy in large models allows interventions targeting spuriously correlated concepts to be compensated by other heads. This redundancy, combined with polysemanticity, helps mitigate conflicts and side effects between properties, as illustrated in Figure 4, where partial head overlaps do not affect unrelated tasks.
>
> **Scaling to more complex concepts [W3, Q4]:** To address the Reviewer’s concern regarding scalability to more complex concepts, we conducted an experiment to verify that our method is still effective when we consider the composition of two simple concepts. This is grounded in the compositionality assumption, recently discussed in the neural network literature [e,f], which posits that complex concepts often emerge from structured compositions of simpler ones. If our method can manipulate compositions of attributes, this offers evidence that it could generalize to more complex and contextual concepts.
>
> We applied inhibitory and enhancing interventions (as in Section 5.2 of the paper) to composite concepts by targeting the union of top-8 heads selected for each of the two base properties. We analyze two property combinations: ‘colors’+‘sentiments’ and ‘colors’+‘quantity’. The two cases differ on the entanglement between the two properties: in the former case, the two sets of heads intersect on one head, while in the latter they are completely disjoint. We report the results of the inhibitory and enhancing interventions in the two settings in the table below:
>
> | Properties (P1+P2)  |Property 1 (inhibitory) $\downarrow$ | Property 1 (enhancing) $\uparrow$| Property 2 (inhibitory) $\downarrow$| Property 2 (enhancing) $\uparrow$|
> |--------------|----------|----------|----------|----------|
> |Colors+Sentiments|0.36|1.23|0.27|3.17|
> |Colors+Quantity|0.22|1.63|0.75|1.16|
>
> These results show that interventions targeting composite head sets yield significant and targeted effects on both underlying properties. This suggests that our method retains functional control even when concepts are more entangled or contextually layered, supporting its potential to scale to more complex scenarios.
>
> **Additional models [W4, Q5]:** Following the Reviewers’ suggestions, we have expanded our model selection, which now includes Gemma3-12B [g], LLaVA-NeXT-13B [h] and Qwen2.5-VL-7B [i]. These include both larger-capacity models and diverse vision-language architectures with strong performance. Results on these models largely confirm trends observed in LLaVA-NeXT-7B, and are reported in detail in the tables below for a selection of image classification and captioning tasks. For each model-task pair we report the results of the most effective intervention among $k={8,16,32}$, normalized with respect to the original model performance, with random baselines (mean over 5 runs) within parentheses. For captioning tasks, we omit CIDEr scores for brevity, as all results exceed 0.91, indicating high-quality generation.
>
> | Model  | SVHN $\downarrow$ | RESISC45 $\downarrow$| DTD $\downarrow$ |Colors (inhibitory) $\downarrow$ | Colors (enhancing) $\uparrow$| Sentiments (inhibitory) $\downarrow$| Sentiments (enhancing) $\uparrow$|
> |--------------|----------|----------|----------|----------|----------|----------|----------|
> |LLaVA-NeXT-13B|0.06 (0.99)|0.55 (0.99)|0.49 (0.94)|0.07 (1.05)|2.74 (1.11)|0.00 (0.97)|7.63 (0.94)|
> |Gemma3-12B|0.13 (0.86)|0.00 (0.83)|0.00 (0.81)|0.33 (0.98)|1.38 (0.87)|0.36 (0.95)|2.06 (1.26)|
> |Qwen2.5-VL-7B|0.01 (0.91)|0.48 (0.65)|0.18 (0.55)|0.14(1.07)|2.84(1.51)|0.94 (1.14)|1.31(0.44)|
>
> We will include complete results for all models in the final version of our paper.
>
> [a] nostalgebraist, interpreting gpt: the logit lens, LessWrong, 2020
>
> [b] Sakarvadia et al., Attention Lens: A Tool for Mechanistically Interpreting the Attention Head Information Retrieval Mechanism, arXiv, 2023
>
> [c] Belrose et al., Eliciting Latent Predictions from Transformers with the Tuned Lens, arXiv, 2023
>
> [d] Dar et al., Analyzing transformers in embedding space, ACL, 2023
>
> [e] Lepori et al., Break It Down: Evidence for Structural Compositionality in Neural Networks, NeurIPS, 2023
>
> [f] Lee et al., Geometric Signatures of Compositionality Across a Language Model's Lifetime, ACL, 2025
>
> [g] Gemma Team, Gemma 3 Technical Report, 2025
>
> [h] Liu et al., LLaVA-NeXT: Improved reasoning, OCR, and world knowledge, 2024
>
> [i] Bai et al., Qwen2.5-VL Technical Report, 2025

---

> > ### Comment · Reviewer_w3Vz · 2025-08-03
> >
> > I thank the authors for their thorough and thoughtful response to my concerns. I have adjusted my score accordingly.
> >
> > My concerns about discussion and comparison with related work have been addressed, as well as those related to architecture generalizability.
> >
> > However, I still believe the impact of the method could be strengthened by evaluating its effectiveness on more abstract concepts, beyond compositions of simpler ones. In addition, a more thorough analysis of the effects of ablating attention heads would be valuable, particularly in downstream tasks involving other concepts that may also rely on the perturbed attention heads, even if those concepts were not the original focus of the analysis.

---

> > > ### Author Response · Authors · 2025-08-04
> > >
> > > We sincerely thank the Reviewer for their response and for taking the time to re-assess their evaluation.
> > >
> > > We are glad that our revisions have addressed their main concerns.
> > >
> > > Regarding the remaining suggestions, we are happy to consider such extensions in the revised version of the paper, if the Reviewer can kindly suggest specific benchmarks or setups that would address these concerns.

---

> > > > ### Comment · Reviewer_w3Vz · 2025-08-08
> > > >
> > > > To include more abstract concepts, you could use items from datasets in which humans rated words along the concrete–abstract dimension, such as those available at https://elexicon.wustl.edu/index.html. It could also be interesting to explore concepts with ambiguous meanings.
> > > >
> > > > Regarding performance maintenance in downstream tasks, I mean verifying that the overall accuracy of the VLM in vision-language tasks (such as image classification, localization, VQA) is not affected by the inhibition of attention heads (at least for concepts that are not targeted).

---

### Official Review · Reviewer_CkpV · 2025-07-01

**Clarity:** 3
**Significance:** 2
**Originality:** 3
**Rating:** 4
**Confidence:** 2

**Summary:**

This paper studies how individual attention heads in text-generative models specialize in specific semantic or visual attributes. Also, they find that editing as few as 1% of the heads, selected using the proposed method, can reliably suppress or enhance targeted concepts in the model output.

**Questions:**

NA

**Ethical Concerns:**

["NO or VERY MINOR ethics concerns only"]

**Final Justification:**

The concerns are addressed in the rebuttal process. I agree with Review #w3Vz that evaluating the proposed method on more concepts can make it more solid.

**Limitations:**

Yes – The authors are forthcoming about the limitations

**Quality:**

3

**Strengths And Weaknesses:**

# Strengths

- Interpretability and Methodological Clarity: The introduction of SOMP to probe attention head specialization offers a transparent and reproducible method for mapping heads to interpretable semantic features. By grounding the analysis in the unembedding matrix, the approach builds a strong conceptual bridge to prevailing interpretability techniques like the logit lens and sparse coding, as clarified in Section 3.

- Thorough Empirical Validation: The method is validated on both LLMs (Mistral-7B) and VLMs (LLaVA-NeXT-7B), with experiments covering textual (question answering, toxicity mitigation) and multimodal (classification, captioning) tasks. For example, Figure 2 and Table 2 provide quantitative results that support the targeted and specific effect of head interventions.

# Weaknesses

- Linearity Assumption and Method Constraints: The core SOMP approach assumes that the behavior of attention heads can be usefully decomposed linearly in the unembedding space (discussed in Section 6). As the authors acknowledge, this may neglect relevant non-linear structures, possibly oversimplifying representational dynamics. For high-capacity transformers, non-linear mixing of semantics is common, and it is unclear how much of a limitation this imposes, especially in deep or multimodal networks.

- Evaluation Scope and Generalization: While the experiments span multiple tasks and domains, the selection of models (Mistral-7B and LLaVA-NeXT-7B) is somewhat narrow. The applicability of findings to much larger or more varied architectures (e.g., GPT-4, PaLM, Flamingo) is not empirically established, nor is generalization to other types of vision-language models.

---

> ### Author Rebuttal · Authors · 2025-07-31
>
> We thank the Reviewer for carefully evaluating our work and for appreciating its methodological foundations. We respond below to the weak points identified in the review.
>
> **W1:** While information flow in transformers is highly nonlinear, it has been shown that large language and vision-language models tend to represent high-level concepts in linear structures in the residual stream [a, b, c, d], especially in late layers close to the output [e].
> This emergent property is the key for the success of our method: even if a concept is expressed in a non-linear structure in some early head, at some stage in the residual stream it will be transformed into a linearly decodable one. We show the validity of this assumption with the empirical evidence provided by our experiments, where our method results in a principled control over the target concept expression at output level.
> We will further clarify this in the Introduction section (L45-L47).
>
>
> **W2:** Following the Reviewers’ suggestions, we have expanded our model selection, which now includes Gemma3-12B [f], LLaVA-NeXT-13B [g] and Qwen2.5-VL-7B [h]. These include both larger-capacity models and diverse vision-language architectures with strong performance. Results on these models largely confirm trends observed in LLaVA-NeXT-7B, and are reported in detail in the tables below for a selection of image classification and captioning tasks. For each model-task pair we report the results of the most effective intervention among $k={8,16,32}$, normalized with respect to the original model performance, with random baselines (mean over 5 runs) within parentheses. For captioning tasks, we omit CIDEr scores for brevity, as all results exceed 0.91, indicating high-quality generation.
>
> | Model  | SVHN $\downarrow$ | RESISC45 $\downarrow$| DTD $\downarrow$ |Colors (inhibitory) $\downarrow$ | Colors (enhancing) $\uparrow$| Sentiments (inhibitory) $\downarrow$| Sentiments (enhancing) $\uparrow$|
> |--------------|----------|----------|----------|----------|----------|----------|----------|
> |LLaVA-NeXT-13B|0.06 (0.99)|0.55 (0.99)|0.49 (0.94)|0.07 (1.05)|2.74 (1.11)|0.00 (0.97)|7.63 (0.94)|
> |Gemma3-12B|0.13 (0.86)|0.00 (0.83)|0.00 (0.81)|0.33 (0.98)|1.38 (0.87)|0.36 (0.95)|2.06 (1.26)|
> |Qwen2.5-VL-7B|0.01 (0.91)|0.48 (0.65)|0.18 (0.55)|0.14(1.07)|2.84(1.51)|0.94 (1.14)|1.31(0.44)|
>
> We will include complete results for all models in the final version of our paper. We note that we did not include proprietary models such as GPT-4, PaLM, or Flamingo in our experiments, as their internal activations and attention head representations are not publicly accessible. Nonetheless, the inclusion of recent, high-performing open-source models at larger scales strengthens our claim that the observed head specialization effects and controllability generalize across architectures.
>
> **References:**
>
> [a] Park et al., The linear representation hypothesis and the geometry of large language models, ICML, 2024
>
> [b] Jiang et al., On the Origins of Linear Representations in Large Language Models, ICML, 2024
>
> [c] Rajaram et al., Line of Sight: On Linear Representations in VLLMs, arXiv, 2025
>
> [d] Papadimitriou et al., Interpreting the linear structure of vision-language model embedding spaces, arXiv, 2025
>
> [e] Gandelsman et al., Interpreting clip's image representation via text-based decomposition, ICLR, 2024
>
> [f] Gemma Team, Gemma 3 Technical Report, 2025
>
> [g] Liu et al., LLaVA-NeXT: Improved reasoning, OCR, and world knowledge, 2024
>
> [h] Bai et al., Qwen2.5-VL Technical Report, 2025

---

> > ### Comment · Reviewer_CkpV · 2025-08-04
> >
> > Thanks for the rebuttal. My concerns are addressed. I have updated my score accordingly.

---

### Official Review · Reviewer_hxZg · 2025-07-03

**Clarity:** 3
**Significance:** 3
**Originality:** 2
**Rating:** 5
**Confidence:** 2

**Summary:**

This paper presents a method to understand and control the internal behavior of transformer models by analyzing individual attention heads that specialize in specific semantic functions. The authors reformulate attention head interpretability using Matching Pursuit, adapting the Simultaneous Orthogonal Matching Pursuit (SOMP) algorithm to identify heads aligned with interpretable concepts such as colours, numbers etc. By projecting attention head outputs into semantic directions derived from the model's unembedding matrix, they reveal a structured and consistent alignment between certain heads and specific output types. They show that certain heads are consistently responsible for specific output types and that manipulating 1% of the heads can enhance/suppress ceratin model behaviours without fine tuning.

**Questions:**

1. How robust is the linear decomposibility assumption?
2. Did you consider alternative semantic bases such as learned concept embedding for representing concepts, why unembedding matrix?
3. How sensituve is the approach to choice and quality of keyword list?
4. How would the intervention effect factual consistency?
5. Are there cases where suppressing a head has no effect because another head compensates for it? How does your method handle this?
Please refer to weakness section for other questions
suggestion : The section on SOMP could benefit from additional intuitive explanation

**Ethical Concerns:**

["NO or VERY MINOR ethics concerns only"]

**Final Justification:**

Thank you for addressing my concerns, my current score reflects my overall impression of the paper!

**Limitations:**

yes

**Quality:**

2

**Strengths And Weaknesses:**

Strength:
1. The paper is well written and easy to follow
2.. Propose a grounded interpretation of attention head interpretability by leevrgaing SOMP and provides meaningful insights into internal workings
3. The method is validated on language and vision language tasks
4. By modifying a very small number of attention heads the model behaviour is steered.
5. The paper has detailed ablation studies to validate the findings

Weakness
1. Using SOMP, the method depends on linear decomposition of attention heads. However transformer dynamics are highly nonlinear, this assumption might over simplify the actual role of attention heads
2. The evaluation is focussed on simple attributes such as colour, numbers etc. It is not clear whether the method would scale to more complex and contextual concepts
3. Concept detection depend on manually curated list pf keywords, this might be incomplete or biased for certain applications.
4. There is no evaluation of hallucination risks after intervention, which is important along with other metrics like semantic drift etc

---

> ### Author Rebuttal · Authors · 2025-07-31
>
> We thank the Reviewer for their insightful assessment of our work and for suggesting several potential improvements to it. We try to address here the main weaknesses and questions that emerged in the review.
>
> **Linear decomposition [W1, Q1]:** While information flow in transformers is highly nonlinear, it has been shown that large language and vision-language models tend to represent high-level concepts in linear structures in the residual stream [a, b, c, d], especially in late layers close to the output [e].
> This emergent property is the key for the success of our method: even if a concept is expressed in a nonlinear structure in some early head, at some stage in the residual stream it will be transformed into a linearly decodable one. We show the validity of this assumption with the empirical evidence provided by our experiments, where our method results in a principled control over the target concept expression at output level.
> We will further clarify this in the Introduction Section (L45-L47).
>
> **Scaling to more complex concepts [W2]:** To address the Reviewer’s concern regarding scalability to more complex concepts, we conducted an experiment to verify that our method is still effective when we consider the composition of two simple concepts. This is grounded in the compositionality assumption, recently discussed in the neural network literature [f, g], which posits that complex concepts often emerge from structured compositions of simpler ones. If our method can manipulate compositions of attributes, this offers evidence that it could generalize to more complex and contextual concepts.
>
> We applied inhibitory and enhancing interventions (as in Section 5.2 of the paper) to composite concepts by targeting the union of top-8 heads selected for each of the two base properties. We analyze two property combinations: ‘colors’+‘sentiments’ and ‘colors’+‘quantity’. The two cases differ in the entanglement between the two properties: in the former case, the two sets of heads intersect on one head, while in the latter they are completely disjoint. We report the results of the inhibitory and enhancing interventions in the two settings in the table below, normalized to the original model performance:
> | Properties (P1+P2)  |Property 1 (inhibitory) $\downarrow$ | Property 1 (enhancing) $\uparrow$| Property 2 (inhibitory) $\downarrow$| Property 2 (enhancing) $\uparrow$|
> |--------------|----------|----------|----------|----------|
> |Colors+Sentiments|0.36|1.23|0.27|3.17|
> |Colors+Quantity|0.22|1.63|0.75|1.16|
>
> These results show that interventions targeting composite head sets yield significant and targeted effects on both underlying properties. This suggests that our method retains functional control even when concepts are more entangled or contextually layered, supporting its potential to scale to more complex scenarios.
>
> **Keyword list quality [W3 and Q3]:** We acknowledge that properties are defined by lists of keywords that may be incomplete or biased. However, as we showed in our experiment on toxicity mitigation (Section 4.2 of the manuscript), our method is able to identify relevant heads over a subset of keywords and then generalize to a disjoint (but semantically related) subset. This suggests that our approach can generalize from a partial or noisy concept definition, as long as the keywords are semantically related to the underlying property. To strengthen this point, in the case of VLMs, we perform a new experiment: we apply our method to identify heads related to 'colors' on Flickr30k (as in Section 5.2), but we replace a fraction of our color-related keywords with random tokens from the model vocabulary. We report in the table below the Jaccard similarity between the original set of top-k heads identified by our method for this property and the top-k heads identified on noisy lists of keywords, while varying the fraction $\rho$ of random keywords:
>
> |k|$\rho=0.05$|$\rho=0.1$| $\rho=0.2$ | $\rho=0.3$ | $\rho=0.4$ | $\rho=0.5$ |
> |----------|----------|----------|----------|----------|----------|----------|
> |16 |0.88| 1.00 | 0.88 | 0.6 | 0.77 | 0.45 |
> |32 |0.83| 0.88 | 0.83 | 0.64 | 0.73 | 0.60 |
>
> These results confirm that head selection is reasonably stable even when up to 50% of the keyword list is replaced with unrelated terms. This robustness suggests that the method is not overly sensitive to noise or incompleteness in the keyword list, and that it can tolerate imperfections in concept specification.
>
> **Hallucination and factual consistency [W4 and Q4]:** We thank the Reviewer for highlighting the importance of evaluating hallucination risks and factual consistency. While we do not explicitly annotate hallucinated content, we address this concern using several well-established automatic metrics that serve as proxies for semantic and factual consistency in our image captioning experiments.
>
> In addition to the CIDEr scores already reported in the submission (which assess how well generated captions match reference captions in content and structure), we now also report BERTScore [h], METEOR [i], and CLIPScore [j]. BERTScore captures fine-grained semantic similarity to reference captions, METEOR reflects syntactic and lexical quality, and CLIPScore directly measures alignment between the generated caption and the image content. We report results in the table below for the strongest intervention setting (32 heads).
> | Metric  |Colors (inhibitory)| Colors (enhancing)| Sentiments (inhibitory)| Sentiments (enhancing)|
> |--------------|----------|----------|----------|----------|
> |METEOR|0.97|0.93|1.00|0.94|
> |BERTScore|1.00|0.99|1.00|1.00|
> |CLIPScore|1.01|0.98|1.00|0.99|
>
> Across both inhibitory and enhancing interventions, all metrics remain stable, indicating that the generated captions preserve semantic coherence, linguistic fluency, and visual factuality. These results suggest that our method does not introduce hallucinations or factual inconsistencies, even when intervening strongly on targeted attributes.
>
> **Concept embeddings [Q2]:** Our work focuses on identifying functionally relevant heads that are responsible for the generation process of specific concepts in VLMs. This motivates the choice of decomposing head representations directly onto the unembedding matrix to identify head properties which are mostly adopted by the model during the generation process. This approach ensures consistency with the model’s decoding process avoiding the introduction of additional alignment overhead.
>
> As correctly pointed out by the Reviewer, there are other approaches for concept discovery. One direction is to use models designed to build representations at a higher semantic abstraction level, such as Large Concept Models [k]. Differently from the unembedding matrix, their representations are not aligned with the token-level representations of LLMs and VLMs, and aligning them would require an additional and possibly complex procedure, especially in the multimodal setting involving text and image tokens. Another approach would be to learn the concept dictionary directly from the head representations, for example extending methods like [l] that analyze representation shifts under fine-tuning. This is a complementary approach, which we acknowledge in the manuscript. However, also these  procedures ultimately require projecting onto the unembedding matrix for interpreting and grounding the concepts.
>
> **Compensating effects [Q5]:** Yes, it is indeed possible that suppressing a single attention head has little or no observable effect due to compensation by other heads. This is a known phenomenon in transformer models, where redundancy across heads can mask the impact of individual interventions. To address this, our method always considers sets of heads rather than individual ones. In practice, intervening on 16 heads is typically sufficient to produce significant effects in a 7B model, as demonstrated across multiple tasks in our experiments. By targeting multiple heads associated with a given property, we mitigate the risk of compensation and capture a broader share of the relevant signal. As a potential extension, one could set a threshold on the cumulative explained variance (rather than fixing the number of heads) to dynamically determine how many and which heads to intervene on. This would allow the method to adapt to the redundancy structure in different models or tasks.
>
> **Intuitive method explanation:**  We thank the Reviewer for the suggestion and agree that a more intuitive explanation would enhance accessibility. We will revise Section 3 in the final version to improve clarity and provide additional intuition for our method.
>
> **References:**
>
> [a] Park et al., The linear representation hypothesis and the geometry of large language models, ICML, 2024
>
> [b] Jiang et al., On the Origins of Linear Representations in Large Language Models, ICML, 2024
>
> [c] Rajaram et al., Line of Sight: On Linear Representations in VLLMs, arXiv, 2025
>
> [d] Papadimitriou et al., Interpreting the linear structure of vision-language model embedding spaces, arXiv, 2025
>
> [e] Gandelsman et al., Interpreting clip's image representation via text-based decomposition, ICLR, 2024
>
> [f] Lepori et al., Break It Down: Evidence for Structural Compositionality in Neural Networks, NeurIPS, 2023
>
> [g] Lee et al., Geometric Signatures of Compositionality Across a Language Model's Lifetime, ACL, 2025
>
> [h] Banerjee and Lavie, METEOR: An automatic metric for MT evaluation with improved correlation with human judgments, IEEvaluation@ACL, 2005
>
> [i] Zhang et al., BERTScore: Evaluating Text Generation with BERT, ICLR, 2020
>
> [j] Hessel et al., CLIPScore: A Reference-free Evaluation Metric for Image Captioning, EMNLP, 2021
>
> [k] Barrault et al., Large Concept Models: Language Modeling in a Sentence Representation Space, arXiv, 2024
>
> [l] Khayatan et al., Analyzing Fine-tuning Representation Shift for Multimodal LLMs Steering, ICCV, 2025

---

### Official Review · Reviewer_WNh7 · 2025-07-15

**Clarity:** 3
**Significance:** 3
**Originality:** 3
**Rating:** 5
**Confidence:** 4

**Summary:**

This paper uses probing to study attention heads in text generative models. They show that they can suppress or enhance targeted concepts generated by the model by editing a small percentage of the heads.

**Questions:**

Why not use Lasso and solve for all the sparse set of coefficients W at the same time? Unlike matching pursuit, which is a greedy algorithm, Lasso will attempt to find the actual optimal global solution.

How does your usage of the unembedding matrix compare to [1] where they also look at using the same matrix as the unembedding matrix, i.e. the weight matrix from the masked language modelling head, to interpret latent / hidden layer outputs.

[1] “Decoding Layer Saliency in Language Transformers” Elizabeth Mary Hou, Gregory David Castanon Proceedings of the 40th International Conference on Machine Learning, PMLR 202:13285-13308, 2023.

**Ethical Concerns:**

["NO or VERY MINOR ethics concerns only"]

**Final Justification:**

The authors' response suitably answered my questions.  I also read the other reviews and rebuttals. And based on all the discussion I have updated my score accordingly.

**Limitations:**

yes

**Quality:**

3

**Strengths And Weaknesses:**

Strengths

They provide a mathematically motivated method for characterizing how individual attention head work.

They are able to that they can degrade performance by inverting important heads versus inverting randomized heads as a base line.

They are able to show good results on a variety of datasets.

Weaknesses

They are only able to show results versus a randomized baseline. There are other papers that claim to also be able to interpret attention heads (including other submissions to this conference). Can you show any comparisons to other works? Being unable to compare against an other works is a critical weakness and fixing this would considerably increase the score.

---

> ### Author Rebuttal · Authors · 2025-07-30
>
> We thank the Reviewer for their feedback on our work, and we respond here to the weaknesses and questions outlined in the review.
>
> **Comparison with baselines:** We agree that including a comparison with prior interpretability methods is a valuable addition to our work. In the absence of any specific request/suggestion from the Reviewer, we adopt the Logit Lens (LL) method [a], which has been vastly employed in interpreting internal representations of LLMs (e.g., [b,c]) and more recently VLMs [d].
>
> In Section 3 of our submission, we already provided a qualitative comparison of our method with LL, showing that Simultaneous Orthogonal Matching Pursuit (SOMP) returns consistently less redundant explanations, due to its inherent orthogonality constraint. Motivated by the Reviewers’ suggestions, to make the comparison quantitative we propose a simple adaptation of LL to head interpretability in VLMs. Specifically, given a keyword-defined property (for example, 'colors'), we assign to each head a score corresponding to the mean logit assigned to property-related tokens after applying LL. The top-k scoring heads are selected, as in our method, and subjected to interventions to evaluate their functional importance.
>
> We report in the table below the results of this analysis on image classification and captioning tasks on LLaVA-NeXT-7B applying the intervention on attention heads. We evaluate performance for k=16 and k=32 attention heads.
>
> |Method      | MNIST $\downarrow$| SVHN $\downarrow$| Eurosat $\downarrow$| RESISC45 $\downarrow$| DTD $\downarrow$|GTSRB $\downarrow$| Colors (inhibitory) $\downarrow$| Colors (enhancing) $\uparrow$| Sentiments (inhibitory) $\downarrow$| Sentiments (enhancing) $\uparrow$|
> |--------------|----------|----------|----------|----------|----------|----------|----------|----------|----------|----------|
> |Ours (k=16)|**0.28**|**0.25**|**0.64**|**0.84**|**1.04**|**0.01**|**0.16**|1.41|**0.04**|**4.97**|
> |Logit Lens (k=16)|0.73|0.27|0.77|1.03|1.05|0.08|0.76|**1.53**|1.13|1.23|
> |Ours (k=32)|**0.12**|**0.17**|**0.42**|0.55|**0.55**|**0.03**|**0.12**|**2.28**|**0.00**|**4.35**|
> |Logit Lens (k=32)|0.14|0.24|0.66|**0.48**|1.02|0.07|0.54|1.19|1.00|0.87|
>
> The results show that our method based on SOMP identifies more functionally relevant heads with respect to LL, as reflected both by stronger decay in performance on image classification tasks and by the stronger effect of concept control interventions (inhibitory and enhancing) in image captioning.
>
> **Q1:** We thank the Reviewer for raising this point. SOMP was chosen specifically because it identifies a shared, sparse support across all examples. This results in a column-sparse matrix $\mathbf{W^*}$, where the same few dictionary atoms (i.e., semantic directions) are active across all samples, and other columns are entirely zero.
>
> In contrast, Lasso would yield a fully sparse matrix with non-zero coefficients scattered across different atoms for different samples, complicating interpretability and head ranking. In the final version of our paper, we will clarify this distinction by explicitly describing $\mathbf{W^*}$ as column-sparse.
>
> **Q2:** We are grateful to the Reviewer for pointing to this related work that we were not aware of. It is true that both our method and [e] rely on the unembedding matrix to interpret intermediate representations. However, the purposes and mechanisms differ, as in [e] the unembedding matrix is used in a gradient-based saliency framework to determine which input tokens are most responsible for classification decisions. The analysis depends on the choice of a specific downstream task and applies techniques inspired by Grad-CAM. In contrast, we use the unembedding matrix as a semantic dictionary to identify which attention heads contribute most to specific latent concepts. Our method is gradient-free, task-agnostic, and operates on head outputs, not input token gradients. In the revised manuscript, we will include a brief discussion of these points, and note that their method could potentially complement ours in refining our analysis on a few, salient tokens.
>
> **References:**
>
> [a] nostalgebraist, interpreting gpt: the logit lens, LessWrong, 2020
>
> [b] Halawi et al., Overthinking the truth: Understanding how language models process false demonstrations, ICLR, 2024
>
> [c] Yu et al., Characterizing Mechanisms for Factual Recall in Language Models, EMNLP, 2023
>
> [d] Neo et al., Towards interpreting visual information processing in vision-language models, ICLR, 2025
>
> [e] Hou and Castanon, Decoding Layer Saliency in Language Transformers, ICML, 2023

---

> ### Comment · Reviewer_WNh7 · 2025-08-04
>
> Thank you for answering my questions. I have updated my score.

---

### Note · Authors · 2025-08-14

We thank all Reviewers for their constructive feedback and the insightful discussion phase, which contributed to a significant improvement of our work. Below, we outline the key strengths of our work as recognized by the Reviewers, along with the main concerns we addressed.


**Core strengths acknowledged:**

-	Mathematically grounded head interpretability method based on Matching Pursuit (WNh7, hxZg), with strong conceptual connections to established methods (CkpV)

-	Transparent and reproducible (CkpV), simple and straightforward to implement (w3Vz)

-	Extensive results on both language and vision-language models (hxZg, CkpV)

-	Broad dataset coverage (WNh7) and detailed ablation studies validating findings (hxZg)

-	Clear explanation (hxZg, w3Vz) and important topic (w3Vz)


**Addressed lack of quantitative comparison (WNh7, w3Vz):**

-	Implemented and evaluated a head-level adaptation of Logit Lens, an established interpretability tool

-	Demonstrated that our method consistently identifies more functionally relevant heads, with stronger and more targeted effects in classification and captioning tasks

-	Expanded discussion of methodological differences vs. Attention Lens and parameter-space analysis, highlighting scalability and applicability to modern VLMs


**Expanded scope & generalization evaluation (hxZg, CkpV, w3Vz):**

-	Added experiments on larger and diverse architectures: Gemma3-12B, LLaVA-NeXT-13B, Qwen2.5-VL-7B

-	Tested on composite semantic concepts, showing scalability to more complex attributes

-	Demonstrated robustness to noisy keyword definitions

-	Confirmed stability and generality of findings across architectures, modalities, and concept structures


**Reviewer-specific follow-ups:**

-	Assessed hallucination/factual consistency (hxZg)

-	Analyzed potential side effects of interventions (w3Vz)

-	Clarified impact of the linearity assumption (CkpV, hxZg)


**Outcome:**

Three Reviewers (WNh7, CkpV, w3Vz) responded to our rebuttal. They acknowledged concerns were addressed and updated their scores.

---

### Decision · Program_Chairs · 2025-09-17

**Decision:**

Accept (spotlight)

**Comment:**

The paper presents a mathematically grounded approach to understand how each attention head in LLM specialize in specific semantic or visual attributes. The approach shows good results steering model outputs with very little edits to the heads (~1%). The reviewers appreciate the importance of topic, clarity and simplicity of the method, the theoretical derivation, extensive empirical results and ablation studies. All reviewers gave accepts or borderline accepts to this paper after the author reviewer discussion period. After considering the reviews and rebuttal, AC  recommends acceptance of the paper